# The Clinical, Microbiological, and Immunological Effects of Probiotic Supplementation on Prevention and Treatment of Periodontal Diseases: A Systematic Review and Meta-Analysis

**DOI:** 10.3390/nu14051036

**Published:** 2022-02-28

**Authors:** Zohre Gheisary, Razi Mahmood, Aparna Harri shivanantham, Juxin Liu, Jessica R. L. Lieffers, Petros Papagerakis, Silvana Papagerakis

**Affiliations:** 1Laboratory of Oral, Head and Neck Cancer—Personalized Diagnostics and Therapeutics, College of Medicine, University of Saskatchewan, 107 Wiggins Road, Saskatoon, SK S7N 5E5, Canada; zog389@usask.ca (Z.G.); razi.mahmood@usask.ca (R.M.); eti908@mail.usask.ca (A.H.s.); 2Department of Mathematics and Statistics, College of Arts and Science, University of Saskatchewan, 106 Wiggins Road, Saskatoon, SK S7N 5E6, Canada; jul086@mail.usask.ca; 3College of Pharmacy and Nutrition, University of Saskatchewan, 107 Wiggins Road, Saskatoon, SK S7N 5E5, Canada; jessica.lieffers@usask.ca; 4Laboratory of Precision Oral Health and Chronobiology, College of Dentistry, University of Saskatchewan, 107 Wiggins Road, Saskatoon, SK S7N 5E5, Canada; petros.papagerakis@usask.ca

**Keywords:** probiotic, periodontal disease, gingivitis, periodontitis, oral health, clinical parameters, prevention, therapeutics

## Abstract

(1) Background: Periodontal diseases are a global health concern. They are multi-stage, progressive inflammatory diseases triggered by the inflammation of the gums in response to periodontopathogens and may lead to the destruction of tooth-supporting structures, tooth loss, and systemic health problems. This systematic review and meta-analysis evaluated the effects of probiotic supplementation on the prevention and treatment of periodontal disease based on the assessment of clinical, microbiological, and immunological outcomes. (2) Methods: This study was registered under PROSPERO (CRD42021249120). Six databases were searched: PubMed, MEDLINE, EMBASE, CINAHL, Web of Science, and Dentistry and Oral Science Source. The meta-analysis assessed the effects of probiotic supplementation on the prevention and treatment of periodontal diseases and reported them using Hedge’s g standardized mean difference (SMD). (3) Results: Of the 1883 articles initially identified, 64 randomized clinical trials were included in this study. The results of this meta-analysis indicated statistically significant improvements after probiotic supplementation in the majority of the clinical outcomes in periodontal disease patients, including the plaque index (SMD = 0.557, 95% CI: 0.228, 0.885), gingival index, SMD = 0.920, 95% CI: 0.426, 1.414), probing pocket depth (SMD = 0.578, 95% CI: 0.365, 0.790), clinical attachment level (SMD = 0.413, 95% CI: 0.262, 0.563), bleeding on probing (SMD = 0.841, 95% CI: 0.479, 1.20), gingival crevicular fluid volume (SMD = 0.568, 95% CI: 0.235, 0.902), reduction in the subgingival periodontopathogen count of *P. gingivalis* (SMD = 0.402, 95% CI: 0.120, 0.685), *F. nucleatum* (SMD = 0.392, 95% CI: 0.127, 0.658), and *T. forsythia* (SMD = 0.341, 95% CI: 0.050, 0.633), and immunological markers MMP-8 (SMD = 0.819, 95% CI: 0.417, 1.221) and IL-6 (SMD = 0.361, 95% CI: 0.079, 0.644). (4) Conclusions: The results of this study suggest that probiotic supplementation improves clinical parameters, and reduces the periodontopathogen load and pro-inflammatory markers in periodontal disease patients. However, we were unable to assess the preventive role of probiotic supplementation due to the paucity of studies. Further clinical studies are needed to determine the efficacy of probiotic supplementation in the prevention of periodontal diseases.

## 1. Introduction

Periodontal disease is a growing public health concern, affecting approximately 750 million individuals worldwide [1]. The burden of this disease is expected to continue to grow as the global population ages [2,3]. Periodontal disease is preventable and reversible in its early stages; however, it can progress to chronic, irreversible states with significant destruction of the tooth-supporting tissues [4]. The cause of periodontal disease is multifactorial with modifiable risk factors, including smoking, unhealthy diet (e.g., a western diet with high sugars and saturated fats), poor oral hygiene, hormonal changes, stress, various medications, and poorly managed comorbidities (e.g., type 2 diabetes), while non-modifiable risk factors include age, sex, and genetics [5]. Periodontal disease, when left untreated, can have local and/or systemic consequences, leading to poor oral and systemic health and quality of life [5,6]. The underlying link of periodontal disease with other chronic systemic diseases likely results from the dissemination of periodontopathogens into the bloodstream, endotoxin release, and the associated imbalanced inflammatory response to periodontopathogens [7,8]. 

Periodontal disease is an inflammatory progressive multi-stage disease of the periodontium (which includes the gingiva, periodontal ligament, alveolar bone, and cementum); this disease is triggered in response to periodontopathogens in the biofilm of the dental plaque on tooth surfaces located near the gingiva (Figure 1) [9,10,11]. The first stage and mildest form of periodontal disease is known as gingivitis [4]. Gingivitis is a reversible condition, and, if untreated, may progress to periodontitis, which is the advanced stage of periodontal disease [4,12,13]. Gingivitis is characterized by redness, swelling, mild irritation and inflammation of the gingival tissue, and mild bleeding on brushing or flossing, while periodontitis is characterized by deep inflammation and loss of alveolar bone and connective tissue between the gingiva and tooth root [9,14]. The progression of periodontal disease is associated with dynamic shifts in the subgingival bacterial counts and composition in the periodontal pocket [15,16,17,18].

### 1.1. Etiology of Periodontal Disease 

The primary etiology of periodontal disease is an imbalanced subgingival microbiome population developing progressively over time due to an increasing relative abundance of periodontal disease-associated bacteria and a corresponding decrease in health-associated bacteria, leading to the disruption of the microbiota–host homeostasis [16,19,20,21]. This gradual phenomenon begins with the early adherence, growth, and colonization of Gram-negative and Gram-positive bacteria on the tooth surface extending sub-gingivally [22]. This provides appropriate conditions for the growth of and colonization by other anaerobic Gram-negative orange and red-complex bacteria [20]. The orange complex consists of *Prevotella intermedia*, *Parvimonas micra*, and *Fusobacterium nucleatum*, while *Porphyromonas gingivalis*, *Tanerella forsythia*, and *Treponema denticola* are components of the red complex [20]. These bacteria are highly pathogenic and have the ability to release bacterial collagenases and other proteases, leading to the stimulation of the pro-inflammatory response and periodontal tissue damage [9,10,20] (Figure 1). 

### 1.2. Periodontal Disease Assessment, Diagnosis, and Therapy

The clinical assessment of periodontal disease includes the evaluation of patient-reported outcomes (i.e., bleeding while brushing or flossing, receding gums, halitosis, sensitive teeth, pain during mastication, and loose teeth) [4] and visual assessment of the distance between the base of the periodontal pocket and gingival margin or cementoenamel junction, the amount of plaque in the gingival margin on the tooth surface, and gingival bleeding [9,23]. All of these factors are measured by standard procedures and defined as specific indexes, including: the plaque index (PlI), gingival index (GI), periodontal pocket depth (PPD), clinical attachment level (CAL), bleeding on probing (BOP), gingival recession (REC), and gingival crevicular fluid (GCF) volume [23,24,25,26,27].

The clinical diagnosis of periodontal diseases is based on the periodontal exam, radiography, and patient’s oral/dental and medical history [28]. The American Academy of Periodontology (AAP) classifies periodontitis into four stages based on severity, complexity, and extent (stage I, II, III, and IV) and three grades based on the evidence of the disease’s progression and its rate (slow, moderate, and rapid) [29,30].

Periodontal therapy consists of the removal of the supra- and subgingival plaque from tooth surfaces using Scaling and Root Planing (SRP) [31,32]. To improve disease outcomes, SRP can be incorporated into surgical procedures or adjunctive antibiotics can be administered [3,31,32,33]. However, antibiotics may cause adverse side effects, or can be contraindicated in some situations; therefore, there is a need for alternative approaches [14].

### 1.3. Periodontal Diseases and Probiotics

Increasing attention has been devoted to probiotic supplementation as a therapeutic adjuvant/alternative to improve oral health [34]. Probiotics are live organisms (usually bacteria) administered to provide health benefits in the prevention or clinical management of different diseases [35,36]. Probiotics have been traditionally accepted in the medical field as an adjuvant treatment of gastrointestinal disorders [37]. Moreover, probiotics are recommended to patients who take antibiotics for the prevention and treatment of Antibiotics-Associated Diarrhea (AAD) [38]. They have also been considered in the clinical management of other conditions, including respiratory tract infections [39]. Furthermore, probiotics may have a therapeutic benefit in dental caries prevention by decreasing the number of cariogenic bacteria, such as *Streptococcus mutans* [40]. Probiotics may function through various mechanisms, including the production of antimicrobial metabolites, immunomodulation, mucosal barrier enhancement, and microbial flora shift through competition for cell adhesion with pathogenic strains [34,41]. The use of probiotics for the clinical management of periodontal diseases is an active area of research. There have been conflicting results based on several individual studies assessing the effects of probiotics on gingival inflammation [42,43]. Furthermore, previously published systematic reviews have had limited findings with conflicting results when examining the clinical efficacy of probiotics on periodontal diseases. For instance, Akram et al. showed no improvement in PlI and GI in patients with gingivitis after probiotic use, while other reviews concluded that probiotics could improve PlI and GI [44,45,46]. A previous systematic review was unable to assess the immunological benefits of probiotic supplementation due to limited studies; however, the included individual studies indicated an immunomodulatory effect of probiotics [47]. Microbiological findings suggested that probiotic supplementation reduced periodontopathogens in subgingival plaque samples [47]; however, there is conflicting evidence in the literature [48]. 

The purpose of our systematic review and meta-analysis was to combine results from randomized clinical trials involving adults with periodontal diseases or healthy volunteers receiving probiotic supplementation (control groups did not receive probiotic supplementation) to assess the effects on the clinical, microbiological, and immunological outcomes related to periodontal disease prevention and management. 

## 2. Materials and Methods

### 2.1. Eligibility Criteria

Studies that were eligible for inclusion in this review satisfied the following criteria: 1. Randomized controlled trials with adults aged 18 years or older clinically diagnosed with either periodontal disease or healthy adults (without periodontal disease); 2. The study design consisted of intervention groups that received probiotics in any form (i.e., lozenge, capsule, tablet, powder, probiotic drink, probiotic-fortified food, toothpaste, mouthwash, spray, or subgingival delivery) and control groups (without probiotic, with a placebo, or with antibiotics); 3. Studies assessing any of the following: clinical, microbiological, or immunological outcomes. Clinical outcomes included: plaque index (PlI), gingival index (GI), probing pocket depth (PPD), clinical attachment level (CAL), bleeding on probing (BOP), gingival recession (REC), and gingival crevicular fluid (GCF) volume. Microbiological outcomes included the subgingival count of periodontopathogens, including: *Porphyromonas gingivalis* (*P. gingivalis*), *Fusobacterium nucleatum* (*F. nucleatum*), *Tannerella forsythia* (*T. forsythia*), *Prevotella intermedia* (*P. intermedia*), and *Aggregatibacter actinomycetemcomitans* (*A. actinomycetemcomitans*), and of commensal oral microbiota, such as *Streptococcus mutans* (*S. mutans*) and *Lactobacillus* species. Immunological outcomes included the GCF levels of matrix metalloproteinase-8 (MMP-8), interleukin-6 (IL-6), interleukin-1β (IL-1β), interleukin-8 (IL-8), interleukin-10 (IL-10), and tumor necrosis factor-α (TNF-α).

English language, peer-reviewed studies published since 2000, which were either open access or accessible to the researchers (via the University of Saskatchewan Library, inter-library loan, or through Google scholar), were included. Detailed information about the inclusion and exclusion criteria are available in Appendix A.

### 2.2. Information Sources, Search Strategy, and Study Selection

Six databases were searched without restrictions, including PubMed, MEDLINE, EMBASE, CINAHL, Web of Science, and Dentistry and Oral Science Source. A combination of keywords and MeSH terms related to the following search domains were used: 1. Periodontal diseases; 2. Clinical, microbiological, and immunological outcomes; 3. Probiotics. These three domains were combined with the “AND” operator. Details of the search strategy are included in the Appendix A.

This systematic review and meta-analysis was registered with the International Prospective Register of Systematic Reviews (PROSPERO) (registration number: CRD42021249120). 

After calibration, two screeners (ZG and AH) independently conducted dual screening (title and abstract, and full-text), and if consensus was not reached, a third author (RM) provided a tie-breaker vote.

### 2.3. Data Items and Collection Process

Two authors (ZG and AH) independently conducted data extraction, then compared the extracted data, and, in case of disagreement, they referred to the publication. The researchers used a Microsoft Excel (Microsoft Inc., Redmond, WA, USA) spreadsheet to record data pertaining to the study design, sample size, age of participants, health status, periodontal disease stage, dose and probiotic strain, treatment and follow-up durations, oral hygiene instructions, mode of probiotic delivery, clinical measurements (PlI, GI, PPD, CAL, BOP, GCF, and REC), oral bacterial count (*P. gingivalis*, *F. nucleatum*, *T. forsythia*, *P. intermedia*, *A. actinomycetemcomitans*, *S. mutans*, and *Lactobacillus species*), immunological outcomes (MMP-8, IL-6, IL-1β, IL-8, IL-10, and TNF-α), and key findings of each study.

### 2.4. Risk of Bias within Studies

The risk of bias within studies was assessed independently by two authors (ZG and AH) using the Cochrane risk–of–bias assessment tool version 2 designed for randomized trials [49]. This tool evaluates within-study bias by assessing the randomization process, deviations from intended interventions, missing outcome data, measurement of the outcome, and selection of the reported result. Studies were categorized as follows: low risk, some concerns, or high risk of bias.

### 2.5. Summary Measures and Synthesis of Results

Comprehensive meta-analysis version 3.3070 (Biostat Inc., Englewood, NJ, USA) was used for the meta-analysis. The meta-analysis was conducted using the mean and standard deviations from the studies at baseline until the immediate follow-up after the end of probiotic treatment. Hedge’s g standardized mean difference (SMD) was calculated with a 95% confidence interval (CI) for each individual study and pooled in the meta-analysis using random-effects models. SMD can be interpreted as follows: a value of 0 indicates that there is no statistically significant difference in the effect of treatment compared to the control; values greater or less than 0 indicate a difference in effect. The strength of the effect can be roughly interpreted as small (SMD = 0.2), medium (SMD = 0.5), and large (SMD = 0.8) [50]. Heterogeneity was assessed using I^2^, which can be interpreted as low (25%), moderate (50%), or high (75%) [51]. 

The meta-analysis used the following steps: 1. Pooled random-effects models examining the effect of probiotic supplementation on clinical, microbiological, and immunological outcomes in periodontal disease patients compared to controls (without probiotics); 2. Pooled random-effects models examining the potential preventive effect of probiotic use on clinical, microbiological, and immunological outcomes in healthy individuals (without periodontal diseases) compared to controls (without probiotics); 3. Subgroup analysis (see Section 2.7). The results of the meta-analysis were reported as the SMD, 95% CI, heterogeneity score (I^2^), and *p*-value. 

Studies lacking data on the mean, standard deviation, and sample size for treatment and/or control were excluded from the meta-analysis, unless these values could be calculated.

### 2.6. Risk of Bias across Studies

Publication bias and small study effects were evaluated using visual inspection of funnel plots and Egger’s regression test. If publication bias/small study effects were detected, we used Duval and Tweedie’s trim and fill methodology to correct for funnel plot asymmetry [52]. 

### 2.7. Additional Analysis (Subgroup Analysis and Investigation of Heterogeneity)

The following subgroup analyses were conducted: 1. Type of periodontal disease (gingivitis, periodontitis); 2. Disease severity; 3. Probiotic treatment duration; 4. Mode of probiotic delivery; 5. Type of probiotic strain; 6. Type of *lactobacillus* species; and 7. Oral hygiene instructions (yes/no). In addition, subgroup analyses assessed the effects of probiotics compared to a control group receiving antibiotics. Disease severity was categorized as moderate (PPD = 4–6 mm) or deep (PPD > 6 mm) periodontal pockets. The treatment duration was categorized as follows: 1. Up to one month; 2. More than one month to two months; or 3. More than two months. The mode of probiotic delivery was categorized as follows: 1. Oral (i.e., toothpaste, mouthwash, sachet applied orally, mouth spray, or oil drops); 2. Oral and ingestion (i.e., lozenges, tablets, or chewing gum); 3. Ingestion (i.e., sachet dissolved in water to drink, capsules, yogurt, or fermented milk/Yakult); or 4. Local application (i.e., subgingival delivery as a paste or gel). The type of probiotic strain was classified as follows: 1. *Lactobacillus* (*Lactobacillus* species only); 2. Mixed (*Lactobacillus* species and other bacterial species); and 3. Other (not *lactobacillus* species). The type of *Lactobacillus* was classified as follows: 1. *Lactobacillus Reuteri*; and 2. Other *Lactobacillus* spp. (any *Lactobacillus* species other than *L. Reuteri*). The oral hygiene instructions were categorized as follows: 1. Yes (when providing specific oral hygiene instructions); or 2. No (when asking participants to maintain their regular oral hygiene habits).

## 3. Results

### 3.1. Study Selection

A total of 1883 publications were identified through the electronic search of the six databases. After removing duplicates, two reviewers conducted title and abstract screening on 994 publications, and 890 were excluded. In the next step, 104 publications underwent full-text screening. Finally, 64 publications remained for the systematic review and 47 were included in the meta-analysis (17 studies were removed due to insufficient/incompatible data for meta-analysis) (Figure 2).

### 3.2. Study Characteristics

There was a total of 64 studies eligible for the systematic review and 47 studies for the meta-analysis. Table 1 presents the detailed individual study characteristics including: periodontal disease status, sample size, probiotic strain, treatment duration and immediate follow-up, mode of probiotic delivery, other treatments, oral hygiene instructions, outcomes investigated, and key findings. The probiotic treatment duration varied from one day to four months. The sample sizes varied from 10 to 120 individuals. The most common probiotic formulation was composed of *Lactobacillus reuteri*. The majority of studies had periodontitis patients.

### 3.3. Risk of Bias within Studies

Using the Cochrane risk–of–bias assessment tool version 2, the majority of studies included were classified as having a low risk of bias (*n* = 38). Additionally, 13 studies were classified as having some concerns, and another 13 studies were classified as having a high risk of bias. The majority of concerns were due to questions related to the randomization process domain. There were minimal concerns regarding the missing outcome data domain. The results are presented in Appendix A.

### 3.4. Synthesis of Results

This meta-analysis used the Hedge’s g standardized mean difference (SMD) to report effect sizes. Forest plots depicting the pooled meta-analysis examining the effects of probiotic supplementation on clinical, microbiological, and immunological outcomes are presented in Figure 3, Figure 4 and Figure 5, respectively. The overall measures of effect are summarized in Appendix A. The results of subgroup analysis are presented in Table 2 with additional information available in the Appendix A. The forest plots for the subgroup analysis of clinical parameters are depicted in Appendix A.

#### 3.4.1. Associations between Probiotic Supplementation and Clinical Outcomes in Periodontal Disease Patients

##### Pooled Meta-Analysis Examining the Effects of Probiotics on Plaque Index (PlI)

The effect of probiotic supplementation on the plaque index (PlI) indicated a statistically significant decrease with probiotic supplementation compared to controls in patients with periodontal diseases (SMD = 0.483, 95% CI: 0.163, 0.803, I^2^ = 67.044, *p*-value ≤ 0.05, and *n* = 13 studies). Upon visual inspection of the funnel plot and confirmation through Egger’s regression test (*p*-value = 0.040), there was evidence of publication bias/small study effects. Duval and Tweedie’s trim and fill method gave an adjusted pooled estimate (SMD = 0.557, 95% CI: 0.228, 0.885, I^2^ = 75.716, *p*-value ≤ 0.05, and *n* = 13 studies) (Figure 3).

##### Pooled Meta-Analysis Examining the Effects of Probiotics on Mean Plaque Percentage (MPP)

A change in the mean plaque percentage (MPP) was reported in 11 studies. The pooled SMD showed a statistically significant decrease in the MPP with probiotic supplementation compared to controls (SMD = 0.879, 95% CI: 0.308, 1.450, I^2^ = 87.544, *p*-value ≤ 0.05, and *n* = 11 studies). There was no evidence of publication bias or small study effects upon visual inspection of the funnel plot and confirmation through Egger’s regression test (*p*-value > 0.05) (Figure 3).

##### Pooled Meta-Analysis Examining the Effects of Probiotics on Gingival Index (GI)

The pooled SMD using data from patients with periodontal disease indicated a statistically significant reduction in the gingival index (GI) with probiotic supplementation compared to controls (SMD = 0.920, 95% CI: 0.426, 1.414, I^2^ = 86.027, *p*-value ≤ 0.05, and *n* = 14 studies). There was no evidence of publication bias or small study effects upon visual inspection of the funnel plot and confirmation through Egger’s regression test (*p*-value > 0.05) (Figure 3).

##### Pooled Meta-Analysis Examining the Effects of Probiotics on Probing Pocket Depth (PPD)

The pooled SMD indicated a statistically significant decrease in the probing pocket depth (PPD) with probiotic supplementation in patients with periodontal disease compared to controls (SMD = 0.578, 95% CI: 0.365, 0.790, I^2^ = 62.840, *p*-value ≤ 0.05, and *n* = 27 studies). There was no evidence of publication bias or small study effect upon visual inspection of the funnel plot, and this was confirmed by Egger’s regression test (*p*-value > 0.05) (Figure 3).

##### Pooled Meta-Analysis Examining the Effects of Probiotics on Clinical Attachment Level (CAL)

The pooled meta-analysis of the effect of probiotic supplementation on the clinical attachment level (CAL) was assessed in patients with periodontitis (*n* = 19 studies). Meta-analysis was not conducted to assess the effects of probiotics on CAL in healthy participants and those with gingivitis due to a lack of studies.

The pooled SMD indicates that there was a statistically significant CAL gain with probiotic supplementation compared to controls in patients with periodontitis (SMD = 0.413, 95% CI: 0.262, 0.563, I^2^ < 0.001, *p*-value ≤ 0.05, and *n* = 19 studies). There was no evidence of publication bias and small study effects upon visual inspection of the funnel plot, which was confirmed by Egger’s regression test (*p*-value > 0.05) (Figure 3).

##### Pooled Meta-Analysis Examining the Effects of Probiotics on Bleeding on Probing (BOP)

The effect of probiotic supplementation on bleeding on probing (BOP) indicated a statistically significant decrease with probiotic supplementation compared to controls in patients with periodontal disease (SMD = 0.756, 95% CI: 0.381, 1.130, I^2^ = 83.699, *p*-value ≤ 0.05, and *n* = 20 studies). Inspection of the funnel plot for potential publication bias or small study effects suggested a right skew. This was confirmed by Egger’s regression test (*p*-value = 0.020). After adjusting with Duval and Tweedie’s trim and fill method, the adjusted SMD remained significant with high heterogeneity (SMD = 0.841, 95% CI: 0.479, 1.20; and I^2^ = 86.750) (Figure 3).

##### Pooled Meta-Analysis Examining the Effects of Probiotics on Gingival Crevicular Fluid (GCF)

The effect of probiotic supplementation on the gingival crevicular fluid (GCF) volume indicated a statistically significant decrease in the GCF with probiotic supplementation compared to controls in patients with periodontal disease (SMD = 0.568, 95% CI: 0.235, 0.902, I^2^ < 0.001, *p*-value ≤ 0.05, and *n* = four studies). There was no evidence of publication bias or small study effects upon visual inspection of the funnel plot and confirmation by Egger’s regression test (*p*-value > 0.05) (Figure 3).

##### Pooled Meta-Analysis Examining the Effects of Probiotics on Gingival Recession (REC)

There was no statistically significant evidence that probiotic supplementation improved gingival recession (REC) compared to controls in patients with periodontitis (*p*-value = 0.741, and *n* = 5 studies). There was no evidence of publication bias or small study effects upon visual inspection of the funnel plot and confirmation by Egger’s regression test (*p*-value > 0.05). 

#### 3.4.2. Associations between Probiotic Supplementation and Microbiological Outcomes in Periodontal Disease Patients

##### Pooled Meta-Analysis Examining the Effects of Probiotics on Subgingival *Porphyromonas Gingivalis* Count

The pooled SMD indicated a statistically significant decrease in the subgingival *P. gingivalis* count with probiotic supplementation compared to controls in patients with periodontal disease (SMD = 0.402, 95% CI: 0.120, 0.685, I^2^ = 10.769, *p*-value ≤ 0.05, and *n* = five studies). There was no evidence of publication bias or small study effects through visual funnel plot inspection and Egger’s regression test (*p*-value > 0.05) (Figure 4).

##### Pooled Meta-Analysis Examining the Effects of Probiotics on Subgingival *Fusobacterium nucleatum* Count

The pooled SMD using data from periodontal disease patients indicated a statistically significant decrease in the subgingival *F. nucleatum* count with probiotic supplementation compared to controls (SMD = 0.392, 95% CI: 0.127, 0.658, I^2^ < 0.001, *p*-value ≤ 0.05, and *n* = 5 studies). There was no evidence of publication bias or small study effects through visual inspection of the funnel plot and Egger’s regression test (*p*-value > 0.05) (Figure 4).

##### Pooled Meta-Analysis Examining the Effects of Probiotic on Subgingival *Tannerella forsythia* Count

The pooled SMD indicated a statistically significant decrease in the subgingival *T. forsythia* count with probiotic supplementation compared to controls in patients with periodontal disease (SMD = 0.341, 95% CI: 0.050, 0.633, I^2^ < 0.001, *p*-value ≤ 0.05, and *n* = four studies). There was no evidence of publication bias or small study effects upon visual inspection of the funnel plot and with Egger’s regression test (*p*-value > 0.05) (Figure 4).

##### Pooled Meta-Analysis Examining the Effects of Probiotics on Subgingival Counts of Other periodonthopathogenes

The meta-analyses conducted to assess the effects of probiotic supplementation on the subgingival bacterial counts of *P*. *intermedia* (*p*-value = 0.193, and *n* = four studies) and *A. actinomycetemcomitans* (*p*-value = 0.164, and *n* = 5 studies) were not statistically significant when compared to controls.

#### 3.4.3. Associations between Probiotic Supplementation and Immunological Outcomes in Periodontal Disease Patients

##### Pooled Meta-Analysis Examining the Effects of Probiotics on Matrix Metalloproteinase-8 (MMP-8) Levels in the Gingival Crevicular Fluid (GCF)

The pooled SMD indicated a statistically significant decrease in the GCF MMP-8 levels with probiotic supplementation compared to the controls in patients with periodontal disease (SMD = 0.819, 95% CI: 0.417, 1.221, I^2^ < 0.001, *p*-value ≤ 0.05, and *n* = two studies) (Figure 5).

##### Pooled Meta-Analysis Examining the Effects of Probiotics on Interleukin-6 (IL-6) Levels in the Gingival Crevicular Fluid

The GCF Interleukin-6 (IL-6) levels showed statistically significant decreases with probiotic supplementation compared to controls in patients with periodontal disease (SMD = 0.361, 95% CI: 0.079, 0.644, I^2^ < 0.001, *p*-value ≤ 0.05, and *n* = three studies). There was no evidence of publication bias or small study effects through visual funnel plot inspection and Egger’s regression test (*p*-value > 0.05) (Figure 5).

##### Pooled Meta-Analysis Examining the Effects of Probiotics on Other Immunological Biomarkers in the Gingival Crevicular Fluid (GCF)

The meta-analyses conducted to assess the effects of probiotic supplementation on GCF levels of IL-1β (*p*-value = 0.393 and *n* = three studies), IL-8 (*p*-value = 0.434 and *n* = two studies), IL-10 (*p*-value = 0.902 and *n* = two studies), and TNF-α (*p*-value = 0.495 and *n* = three studies) in periodontal disease patients were not statistically significant when compared to controls.

## 4. Discussion

The aim of this systematic review and meta-analysis was to examine if probiotic supplementation is associated with preventive and therapeutic benefits in terms of improvement of clinical, microbiological, and immunological outcomes in patients with periodontal disease. In summary, the findings of this analysis are promising and indicate that probiotic supplementation significantly improved clinical outcomes, decreased certain periodontopathogen counts, and reduced the levels of specific pro-inflammatory biomarkers in patients with periodontal disease. 

The overall results of this meta-analysis showed statistically significant improvements in all clinical parameters (PlI, MPP, GI, PPD, CAL, BOP, and GCF volume) in patients with periodontal disease after probiotic supplementation compared to control groups who did not receive probiotics. When examining the microbiological outcomes, the subgingival periodontopathogen counts of *P. gingivalis*, *F. nucleatum*, and *T. forsythia* showed statistically significant reductions with probiotic supplementation. Furthermore, there was a statistically significant reduction in the GCF levels of inflammatory mediators (MMP-8 and IL-6) with probiotic supplementation.

In addition to the pooled meta-analysis, the effects of probiotic supplementation on clinical outcomes were further assessed through subgroup analysis based on:Type of periodontal disease, which indicated that probiotic supplementation improved clinical outcomes in patients with periodontitis, but not in those with gingivitis or healthy individuals. However, the GCF volume had statistically significant reductions in both gingivitis and periodontitis patients;Probiotic formulations consisting of *Lactobacillus* species and, more specifically, *L. reuteri* were associated with statistically significant improvements in all clinical outcomes in patients with periodontal disease;Probiotic treatment duration, which showed that probiotic supplementation resulted in statistically significant improvements in the clinical outcomes after one month of supplementation in periodontal disease patients;Mode of probiotic delivery, which indicated that probiotic supplementation through the “oral and ingestion” mode was associated with statistically significant improvements in all clinical outcomes in periodontal disease patients.Oral hygiene instructions along with probiotic supplementation improved PlI and BOP, while GI, PPD, and CAL improved with probiotic supplementation, irrespective of the presence or absence of oral hygiene instructions.

Periodontal disease stability refers to a state of successful treatment where clinical indicators do not progress in severity and is characterized by a minimized BOP score and improved PPD and CAL scores [111,112]. The results of our meta-analysis suggest that probiotic supplementation is a promising adjuvant to the current standard of care (SRP) to improve prognosis and clinical outcomes in patients with periodontitis. Given the improvements in the clinical, immunological, and microbiological outcomes in periodontal disease patients, it seems plausible that probiotics may contribute to reaching periodontal stability.

### 4.1. Probiotics and Severity of Periodontal Disease

Our subgroup analysis did not have significant findings in gingivitis patients, which may be partially attributed to: 1. Gingivitis being an early stage of periodontal disease, and, therefore, the clinical aspects of disease and consequently the effects of probiotic supplementation may be less notable; 2. Some studies on gingivitis did not use probiotics along with the standard of care; and 3. These results could be due to the lack of eligible studies and the moderate to high levels of heterogeneity in our analysis. However, the statistically significant improvements that our meta-analysis found in all clinical parameters at the late/severe disease stages (periodontitis) adds evidence to the existing literature on the clinical benefits of probiotic supplementation in the management of periodontal disease.

### 4.2. Probiotics vs. Antibiotics

The current non-surgical standard of care for periodontal disease includes SRP, which is sometimes followed by antibiotic treatment [31,33]. However, previous studies have reported that antibiotic use, particularly long-term administration, may increase the risk of oral and gut dysbiosis, antibiotic-resistant bacteria, allergic reactions, and other side effects [42]. Probiotics are a promising adjuvant/alternative approach that can be taken safely for longer periods of time. Our meta-analysis comparing probiotic supplementation to antibiotics as adjuvants to SRP did not show any statistically significant differences in CAL and PPD in periodontitis patients. (Note: The other clinical outcomes were not assessed due to insufficient studies). This finding suggests that probiotic supplementationmay be as effective as antibiotics in improving clinical outcomes. Previous studies found similar results when either one is administered as an adjuvant to SRP [61,83]. Furthermore, a more recent study concluded that probiotics are a more effective adjuvant with SRP than antibiotics [113]. However, there is evidence indicating that, when probiotics or antibiotics were administered independently, neither one was more efficacious than SRP alone [10,61]. Although these conclusions warrant more research, our meta-analysis indicated improved clinical outcomes, and it is plausible from a mechanistic perspective that probiotic supplementation as an adjuvant to SRP may be a safer and more effective long-term therapeutic option in the management of periodontal disease compared to antibiotics.

### 4.3. Probiotic Formulation and Duration

Our findings also indicated that probiotic formulation and the type of probiotic strain have clinical relevance. Specifically, our subgroup analysis examining the type of probiotic strain indicated that *Lactobacillus* probiotic formulations improved all clinical outcomes. Further analysis into the types of *Lactobacillus* species identified that *L. reuteri-*containing probiotic formulations improved all clinical outcomes. *Lactobacillus* species may play a role in plaque control through direct interaction via colonization resistance, which includes competition for binding sites and nutrients or antimicrobial agent production to inhibit other oral bacteria [114]. Additional evidence suggests that *L. reuteri* may also influence the composition of the oral microbiome and is correlated with reduced periodontopathogens, including *F. nucleatum* [115,116]. As oral microbiome dysbiosis is one of the root causes of periodontal disease, the shift induction by *L. reuteri* may result in improvements in periodontal disease clinical outcomes. The only study without *Lactobacillus* included in our meta-analysis with statistically significant reduction of PPD and CAL, but not PlI, used the subgingival delivery of *Saccharomyces Boulardii* [58]. *S. Bouldarii* is a yeast probiotic that is non-colonizing [117], which likely caused nonsignificant changes in plaque accumulation [58]. However, the improvements in other clinical parameters (PPD and CAL) may be due to its role in immunomodulation [118].

Two out of eleven studies with *L. reuteri* probiotic formulations in patients with periodontitis did not result in significant reductions in plaque accumulation [62,119]. Elsadek et al. recruited periodontitis patients with type 2 diabetes, and the most immediate follow-up data available for our analysis was nine weeks after ending the probiotic treatment [62]. Non-significant plaque reduction could be due to the fast turnover of the oral microbiome after probiotic treatment completion [116], although a recent meta-analysis conducted to assess the long-term efficacy of probiotics found that clinical improvements persisted for up to eleven months after the completion of probiotic treatment [47]. Our subgroup analysis based on probiotic treatment duration suggested that the beneficial effects of probiotics on clinical outcomes may occur within a month. However, the optimal treatment duration cannot be inferred from our meta-analysis because there were few studies of participants with a treatment duration of more than one month to two months. 

### 4.4. Probiotic Mode of Delivery

Studies have suggested that probiotics may have different mechanisms of action to improve periodontal disease, including direct interaction with oral bacteria and/or indirect action in the oral cavity through the modulation of innate and adaptive immunity [114] and the gut–oral microbiome axis [120]. It has also been suggested that probiotics need to have enough contact time with the oral environment to become part of the biofilm and fight against pathogens through competition for binding sites, nutrients, and the production of anti-microbial substances [114,121]. Probiotics can also interact with immune cells, altering the production of inflammatory mediators, thereby modulating the immune response in the oral cavity [114]. Probiotics are also often administered to balance the gut microbiome [122]. A well-balanced gut microbiome regulates overall immune homeostasis [123,124]. The high count of periodontopathogens in periodontal disease may translocate to the gut resulting in gut microbiome dysbiosis [120]. Thus, it is plausible that probiotic supplementation may improve periodontal disease severity, at least in part, via this indirect mechanism linking the gut and oral microbiomes. Therefore, different modes of probiotic delivery may improve clinical outcomes based on the type of administration (oral, ingestion, or a combination of both). Interestingly, our meta-analysis indicated that probiotic administration via ‘ingestion’ only improved signs of tissue destruction (PPD and CAL). However, all clinical outcomes showed improvements with ‘oral and ingestion’ probiotic administration, potentially due to the combined direct and indirect mechanisms of action.

### 4.5. Probiotic Supplementation and Oral Hygiene Instructions

Poor oral hygiene leads to periodontal disease progression, recurrence, and treatment failure [125]. Oral hygiene instructions (OHI) play a key role in all stages of periodontal disease management. Educating patients on the importance of oral hygiene practices and techniques to ensure motivation to adhere to oral hygiene routines are critical components of OHI [125]. Our analysis showed that, irrespective of the oral hygiene instructions, probiotic supplementation improved GI, PPD, and CAL, but not plaque accumulation. Poor plaque control is a key determinant of periodontal disease [126]. Effective OHI can control and reduce plaque accumulation and improve the gingival health status, independent of professional prophylactic care [127,128]. Supporting this evidence, our findings indicated that probiotic supplementation combined with OHI resulted in statistically significant improvements in all clinical outcomes, including plaque accumulation. This suggests that probiotics may synergistically improve clinical outcomes with OHI, stressing the importance of OHI in effectively preventing/reducing plaque accumulation.

### 4.6. Probiotic Supplementation and GCF Volume

In the healthy gingival sulcus, the GCF flow and volume are lower compared to those of the inflamed gingival sulcus [25]. Increased GCF flow is a host defense mechanism to flush bacteria and their metabolites from the gingival sulcus [25]. Furthermore, the GCF content, including the microbial composition and concentration of inflammatory mediators, differs between periodontal disease and healthy individuals, as well as during disease progression and during and after treatment [129]. This is in line with findings from a study by Teles et al., who found a greater GCF volume and inflammatory mediators when comparing clinically healthy oral sites in periodontitis patients compared to healthy individuals. Their results suggested a higher risk of periodontal disease initiation and progression from healthy sites in patients with periodontitis [130]. Altogether, these findings indicate the potential clinical importance of the GCF volume and its level of inflammatory mediators as an early diagnostic test for periodontal disease prior to the onset of clinical manifestations. Based on our findings, the GCF volume was significantly decreased after probiotic supplementation in both gingivitis and periodontitis patients, and other clinical outcomes also improved in these patients [67,72,74,106]. Although our study did not find statistically significant differences between probiotic supplementation and GCF volume in healthy participants, it should be noted that each individual study included in our analysis reported that the GCF volume increased without probiotic supplementation, but not with probiotic supplementation [42,78]. Probiotics may potentially be of therapeutic value for the prevention and early treatment of periodontal disease. Due to the limited number of studies included in our analysis, our results need to be interpreted with caution until further studies fully ascertain the preventive role of probiotics in periodontal disease.

### 4.7. Probiotic Supplementation and Microbiological Outcomes

Our meta-analysis examined the effects of probiotic supplementation on subgingival bacterial counts in periodontal disease. Our results found statistically significant reductions in the periodontopathogens *P. gingivalis* and *T. forsythia* from the red complex and *F. nucleatum* from the orange complex after probiotic supplementation. Teles et al. reported that subgingival plaque samples from healthy sites in patients with periodontitis had a higher proportion of orange and red-complex bacteria compared to healthy individuals (without periodontitis), suggesting a higher risk for these sites to be colonized by periodontopathogens [130]. Hence, probiotics, by reducing periodontopathogens, may be beneficial in inhibiting the progression of periodontal diseases to healthy sites. Orange-complex bacteria create favorable conditions for colonization by red-complex bacteria, which are strictly anaerobic [131]. *F. nucleatum* plays a crucial role as a bridging organism between early and late colonizers, including red-complex bacteria [132]. Red-complex pathogens are higher in number in plaques found in deeper periodontal pockets and advanced lesions [20]. Although, in our meta-analysis, probiotic supplementation resulted in statistically significant improvements in PPD for both moderate and deep pockets, this improvement was more pronounced in deeper pockets. These findings and their potential ramifications may imply that probiotic supplementation has the potential to alter the relative abundance of orange and red-complex bacteria, thus reducing oral tissue destruction and leading to conditions favoring periodontal healing/stability.

### 4.8. Probiotic Supplementation and Immunological Outcomes

Our analysis of the effects of probiotic supplementation on immunological outcomes indicated that the GCF levels of MMP-8 and IL-6 were reduced after probiotic supplementation in patients with periodontal disease. Previously, MMP-8 and IL-6 were identified as salivary biomarkers that increased in periodontitis [133,134]. MMP-8 is the main collagenase involved in periodontal disease, with the highest collagenolytic activity in GCF [135]. A recent meta-analysis reported significantly higher levels of salivary MMP-8 in periodontitis patients compared to healthy participants [135]. Similarly, a study found that the GCF in the healthy sites of periodontitis patients had elevated inflammatory mediators, including MMP-8, when compared to healthy individuals [130], suggesting that inflammatory mechanisms may occur before they are clinically diagnosed or symptomatic. Moreover, additional evidence indicates that higher GCF levels of MMP-8 may predict periodontal disease progression [136], and that MMP-8 can also be measured as a grading biomarker to classify disease stage and progression in periodontitis [137]. 

IL-6 is an inflammatory mediator induced by pathogens and other pro-inflammatory cytokines, and has been reported to be elevated in periodontitis [138]. A recent meta-analysis concluded that the GCF levels of IL-6 are significantly increased in patients with chronic periodontitis [139]. However, there is conflicting evidence in the literature, with other studies suggesting that there is no correlation between the IL-6 levels and periodontal disease [140,141]. The proposed role of IL-6 in periodontal disease pathogenesis is through the stimulation of MMP production and the activation of pathways involved in inflammation [142]. Accordingly, there may be a link between the IL-6 levels and clinical parameters of periodontal disease. The pooled results of our meta-analysis showed statistically significant reductions in the GCF levels of IL-6 after probiotic supplementation, despite individual studies having differing results. Twetman et al. indicated that the use of *L. reuteri* probiotic gum reduced the GCF levels of IL-6 and numbers of sites with bleeding on probing [106]. Keller et al. did not observe any significant change in the GCF levels of IL-6 with other strains of *Lactobacillus* probiotics; however, BOP improved significantly compared to the baseline in both the probiotic and control groups [72]. Shetty et al. found a significant reduction in the GCF levels of IL-6 after non-*Lactobacillus*-containing probiotic supplementation in periodontitis patients compared to controls, but not clinical outcomes [95]. Further research is required to determine the potential effects of different probiotic formulations on inflammatory molecular pathways and their impact on the clinical aspects of periodontal diseases. 

### 4.9. Probiotic Supplementation and Periodontal Disease Prevention

When examining the potential preventive role of probiotic supplementation in periodontal disease in healthy individuals, the results of our meta-analysis did not find any statistically significant improvements in PlI and GI after probiotic supplementation. There were insufficient studies to analyze the effects of probiotic supplementation on other outcomes in healthy participants to clearly assess the preventive potential of probiotics in periodontal diseases. 

### 4.10. Strengths and Limitations

This is the first study examining the influence of probiotic formulation, mode of delivery, treatment duration, and the impact of oral hygiene instructions on periodontal diseases using subgroup analysis. Another strength of this study is that it used an up–to–date analysis of the literature, used a systematic approach, and examined emerging factors that may assist in the prevention, diagnosis, and treatment of periodontal diseases.

A limitation of this meta-analysis was that the number of studies examining microbiological and immunological outcomes was scarce; thus, some of the results need to be interpreted with caution. We used Hedge’s g standardized mean difference to provide better accounting for small study sample sizes. A further limitation was that, due to the aggregated data, we were unable to examine the effects of probiotic supplementation on periodontal diseases stratified by risk factors, such as sex, age, smoking status, and comorbidities. We were also unable to investigate the correlation between periodontopathogens and inflammatory biomarkers before and after probiotic treatment.

## 5. Conclusions

This systematic review and meta-analysis highlights the potential therapeutic benefits of probiotic supplementation in the treatment of periodontal disease. The results indicate that probiotic supplementation improved the clinical parameters, reduced the subgingival bacterial counts of specific periodontopathogens, and reduced the GCF levels of some proinflammatory mediators in periodontal disease patients. The impact of probiotic supplementation on clinical outcomes is affected by the probiotic formulation, mode of delivery, treatment duration, and the type of periodontal disease. More research is needed to better assess the therapeutic and preventive value of probiotic supplementation in patients with gingivitis (early disease), as well as in healthy (without periodontal disease) individuals.

## Figures and Tables

**Figure 1 nutrients-14-01036-f001:**
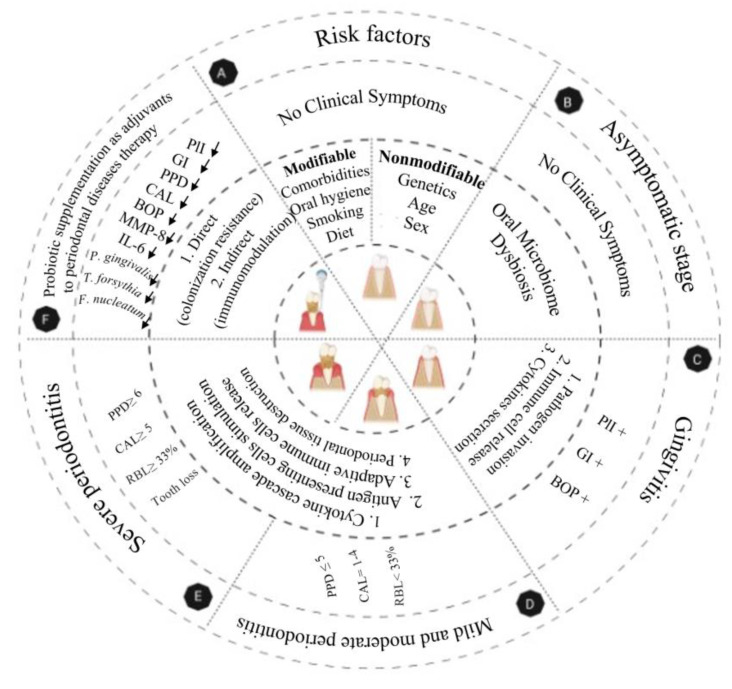
Etiology and pathogenesis of periodontal diseases. Periodontal disease is initiated by disrupting the commensal oral microbiome–host homeostasis. (**A**). Both modifiable and nonmodifiable risk factors impact the oral microbiome composition and disrupt homeostasis between the host and microbiome. Modifiable risk factors include diet, smoking, oral hygiene, and comorbidities (such as type 2 diabetes), while genetics, age, and sex are nonmodifiable risk factors. (**B**). Disrupted homeostasis provides appropriate conditions for the growth of periodontopathogens and biofilm formation on the tooth surfaces extending sub-gingivally. There are no clinical symptoms in this stage. (**C**). These bacteria penetrate and grow in the gingival epithelium. Host–bacteria interactions cause a chemotactic gradient that attracts innate immune cells, including neutrophils, macrophages, and NK cells, to the affected sites. In addition, the outgrowth of bacteria progressively destroys the tissue and provides enough nutrients for more pathogen growth, followed by increased activity of innate immune cells and the secretion of pro-inflammatory cytokines, including IL-1, IL-8, and TNF. Early clinical symptoms in this stage are redness, swelling, mild inflammation, and bleeding of the gingiva, which are diagnosed by measuring the PlI, GI, and BOP. (**D**). Then, Antigen-Presenting Cells (APC), including dendritic cells, present bacterial antigens to lymphocytes and trigger adaptive immune system activity and antibody and cell-mediated immune responses, resulting in a pro-inflammatory response with high expression of IL-4, 6, 8, 10, 12, TGF-β, and IFN-γ. (**E**). High levels of these inflammatory mediators stimulate more inflammatory mediators, causing periodontal tissue destruction and leading to the loss of the gingival attachment to the tooth, and causing deep pockets around the teeth that provide appropriate conditions for the growth and colonization of other anaerobic periodontopathogens. Untreated, these pathophysiological changes can lead to alveolar bone resorption and, ultimately, tooth loss in the most advanced stage of the disease. (**F**). Probiotics may have therapeutic benefits in periodontal disease treatment when used as an adjuvant to standard periodontal care. Various mechanisms of action have been considered for the role of probiotics in periodontal disease improvement. Probiotics interact directly with periodontopathogens through colonization resistance, which includes competition for binding sites and nutrients, and the production of antibacterial agents inhibiting pathogen growth. Probiotics can play a role in periodontal disease improvement indirectly via the modulation of innate and adaptive immunity and through the gut–oral microbiome axis. PlI, Plaque Index; GI, Gingival Index; BOP, Bleeding on Probing; IL, Interleukin; TGF-β, Tumor Growth Factor-β; and IFN-γ, Interferon-γ.

**Figure 2 nutrients-14-01036-f002:**
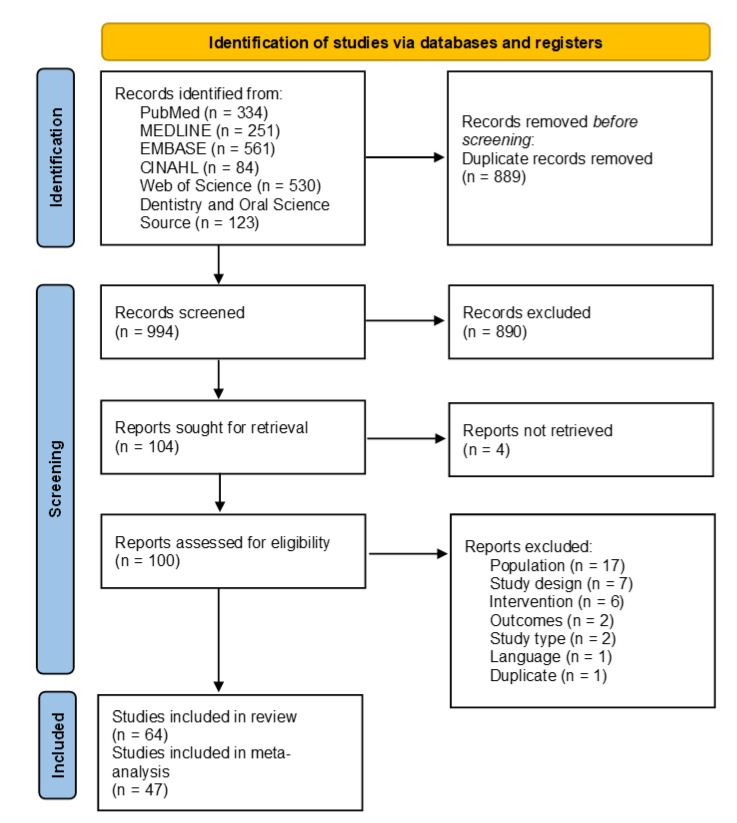
Preferred Reporting Items for Systematic Reviews and Meta-Analyses (PRISMA) flow diagram detailing the study selection.

**Figure 3 nutrients-14-01036-f003:**
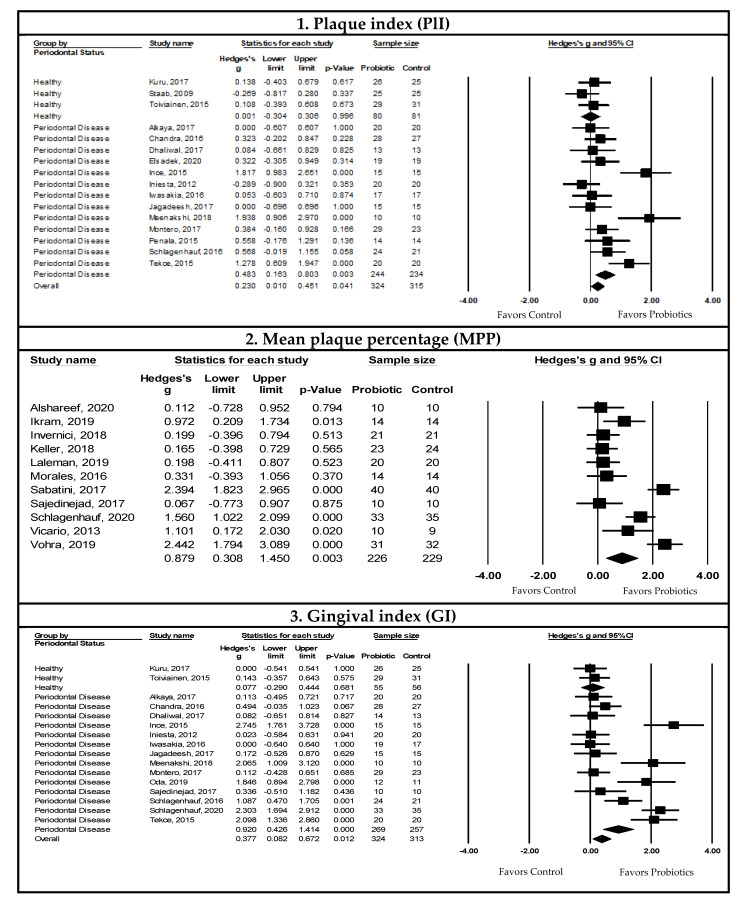
Pooled meta-analysis examining the effects of probiotic supplementation on clinical outcomes. 1. Forest plot of the Hedge’s g SMD comparing the effects of probiotic supplementation to control groups on the plaque index (P1I) using a random-effects model. Note that we detected publication bias and/or small study effects, and the adjusted Hedge’s g SMD = 0.557, 95% CI: 0.228, 0.885, and *p*-value ≤ 0.05 [14,41,43,53,58,60,62,67,69,70,80,86,91,105]. 2. Forest plot of the Hedge’s g SMD comparing the effects of probiotic supplementation to control groups on the mean plaque percentage (MPP) using a random-effects model [54,66,68,72,77,82,88,89,92,107,109]. 3. Forest plot of the Hedge’s g SMD comparing the effects of probiotic supplementation to control groups on the gingival index (GI) using a random-effects model [14,41,43,53,58,60,69,70,75,80,89,91,92,105,110]. 4. Forest plot of the Hedge’s g SMD comparing the effects of probiotic supplementation to control groups on the probing pocket depth (PPD) using a random-effects model [10,34,41,53,54,55,58,60,62,64,66,67,68,69,74,75,76,77,80,81,82,85,89,92,102,103,109,110]. 5. Forest plot of the Hedge’s g SMD comparing the effects of probiotic supplementation to control groups on the clinical attachment level (CAL) using a random-effects model [10,34,54,58,62,66,68,76,77,80,82,85,92,102,103,109]. 6. Forest plot of the Hedge’s g SMD comparing the effects of probiotic supplementation to control groups on bleeding on probing (BOP) using a random-effects model. Note that we detected publication bias and/or small study effects, and the adjusted Hedge’s g SMD = 0.841, 95% CI: 0.479, 1.200, and *p*-value ≤ 0.05 [10,34,41,53,56,66,67,69,70,72,74,76,77,88,89,102,103,107,109]. 7. Forest plot of the Hedge’s g SMD comparing the effects of probiotic supplementation to control groups on the gingival crevicular fluid (GCF) using a random-effects model [42,67,72,74,75,106].

**Figure 4 nutrients-14-01036-f004:**
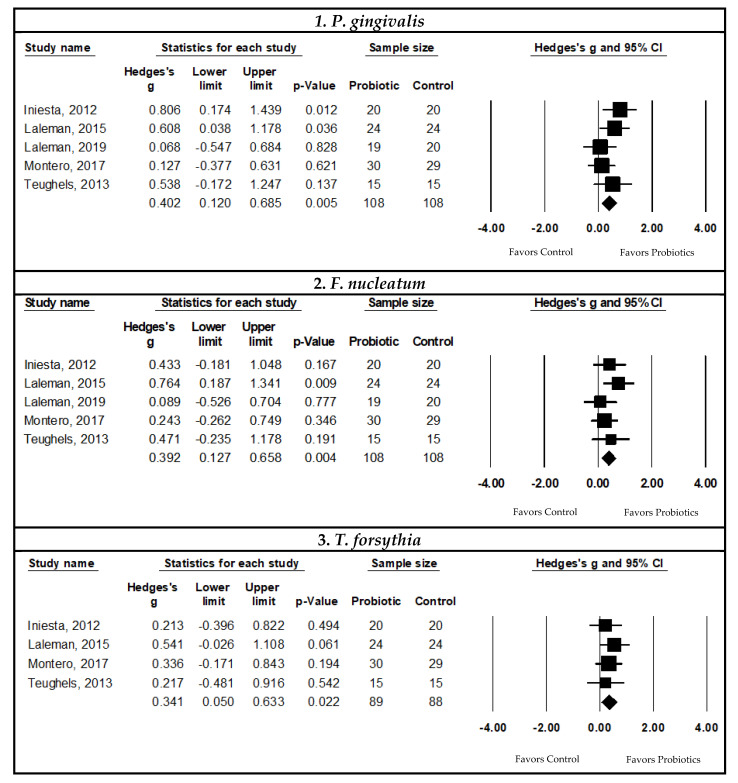
Pooled meta-analysis examining the effects of probiotic supplementation on microbiological outcomes in periodontal disease patients. 1. Forest plot of the Hedge’s g SMD comparing the effects of probiotic supplementation to control groups on the subgingival *P. gingivalis* bacterial count using a random-effects model [14,43,76,77,102]. 2. Forest plot of the Hedge’s g SMD comparing the effects of probiotic supplementation to control groups on the subgingival *F. nucleatum* bacterial count using a random-effects model [14,43,76,77,102]. 3. Forest plot of the Hedge’s g SMD comparing the effects of probiotic supplementation to control groups on the subgingival *T. forsythia* bacterial count using a random-effects model [14,43,76,102].

**Figure 5 nutrients-14-01036-f005:**
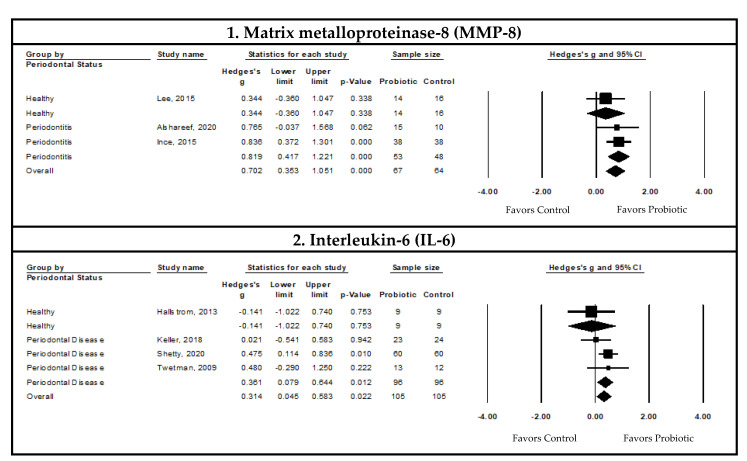
Pooled meta-analysis examining the effects of probiotic supplementation on immunological outcomes in periodontal disease patients; 1. Forest plot of the Hedge’s g SMD comparing the effects of probiotic supplementation to control groups on the gingival crevicular fluid (GCF) level of matrix metalloproteinase-8 (MMP-8) using a random-effects model [59,67,78]. 2. Forest plot of the Hedge’s g SMD comparing the effects of probiotic supplementation to control groups on the GCF level of interleukin-6 (IL-6) using a random-effects model [42,72,95,106].

**Table 1 nutrients-14-01036-t001:** Characteristics of the studies included in the systematic review and meta-analysis.

Author, Year, Country	Study Sample Characteristics			Probiotic Formulation	Treatment Duration/ Immediate Follow-Up	Mode of Probiotic Delivery	Other Treatments	Oral Hygiene Instructions	Outcomes Investigated	Key Findings
	Disease Status	Sample Size								
		Probiotic	Control							
Alkaya, 2016 [53] Turkey	Gingivitis	20	20	*B. megaterium*, *B. pumulus*, *B. subtilis*	8 weeks/ 8 weeks	Toothpaste, mouth rinse, and toothbrush	Supragingival scaling and/or oral prophylaxis	Yes	PlI *, GI *, PPD *, BOP *	No statistically significant difference attributed to probiotic use in gingivitis patients.
Alshareef, 2020 [54] Saudi Arabia	Periodontitis	15	10	*B. bifidum*, *L. acidophilus*, *L. casei*, *L. rhamnosus*, *L. salivarius*	30 days/ 30 days	Lozenge	SRP	Yes	PlI *, CAL *, PPD *, GBI ^+^, GCF *, MMP-8 *	Statistically significant improvement in GBI and greater improvement in GCF with probiotic use.
Bazyar, 2020 [55] Iran	Periodontitis	23	24	*Bifidobacterium*, *B. longum*, *L. acidophilus*, *L. bulgaricus*, *L. casei*, *L. rhamnosus*, *S. thermophilus*	8 weeks/ 8 weeks	Capsule	NSPT	No	PlI ^+^, CAL^+^, BOP ^+^, PPD *, IL1β ^+^, MDA ^+^, TAC ^+^, SOD *, CAT, GPx ^+^	Probiotic supplementation and NSPT in type 2 diabetes patients with chronic periodontitis may improve antioxidant, anti-inflammatory, and periodontal parameters.
Bollero, 2017 [56] Italy	Gingivitis	19	21	*B. animalis*, *B. bifidum*, *L. acidophilus*, *L. delbrueckii*, *L. plantarum*, *L. reuteri*, *L. lactis*, *S. thermophilus*	1 week/ 1 week	Mouthwash	None	Not mentioned	BOP ^+^, PCR ^+^	Probiotic mouthwash may serve as an additional prophylactic to standard oral hygiene procedures.
Boyeena, 2019 [57] India	Periodontitis	10	10	*B. bifidum*, *B. longum*, *L. acidophilus*, *L. rhamnosus*	Once/ 45 days	Paste	1) SRP + tetracycline fibers 2) SRP + tetracycline fibers + Probiotic	Yes	PlI *, PPD ^+^, SBI ^+^, total bacteria *	Probiotic and tetracycline may act synergistically in the treatment of periodontitis.
Chandra, 2016 [58] India	Periodontitis	28	27	*S. boulardii*	Once/ 1 week	Paste	SRP	Yes	PI, MGI *, CAL ^+^, PPD ^+^	*S. boulardii* and SRP significantly improved periodontal disease parameters compared to SRP alone.
Deshmukh, 2017 [59] India	Healthy	15	15	*Bifidobacterium*, *Lactobacillus*, *S. Boulardii*	14 days/ 14 days	Sachet	Supragingival scaling + chlorhexidine mouthwash control	Yes	PlI *, GI *	Probiotic mouthwashes have similar efficacy to chlorhexidine and are a potential alternative with fewer side effects.
Dhaliwal, 2017 [60] India	Periodontitis	14	13	*B. mesentericus*, *C. butyricum*, *L. sporogenes*, *S. faecalis*	21 days/ 30 days	Lozenge	SRP	Not mentioned	PI *, GI *, PPD *, RAL *, *A. actinomycetemcomitans*, *P. gingivalis ^+^*, *P. intermedia*	Probiotics may be used as an adjunctive treatment for the management of chronic periodontitis.
Duarte, 2019 [61] United Arab Emirates	Gingivitis	5	5	*S. oralis*, *S. rattus*, *S. uberis*	30 days/ 30 days	Mouthwash	1) SRP 2) SRP + chlorhexidine mouthwash	Yes	GI *, OHI ^+^, PI *	Changes may be attributed to type and duration of intervention. Probiotics showed similar efficacy to chlorhexidine and better results compared to SRP alone.
Elsadek, 2020 [62] Saudi Arabia	Periodontitis	19	19	*L. reuteri*	3 weeks/ 12 weeks	Lozenge	1) RSD + Photodynamic therapy 2) RSD alone	Yes	CAL *, BOP *, PPD *, PS*, *P. gingivalis* ^+^, *T. Forsythia* ^+^, *T. denticola ^+^*	Photodynamic therapy showed greater benefits for deeper periodontal pockets. Probiotics reduced bacterial counts more than RSD alone.
Ercan, 2020 [63] Turkey	Gingivitis	40	40	*B. lactis*, *B. longum*, *E. faecium*, *L. acidophilus*, *L. plantarum*, *S. thermophilus*,	1 month/ 1 month	Chewing tablet	SRP	Yes	PlI^+^, GI *, GCF *, IL-6 ^+^, IL8 ^+^, IL10 ^+^	Adjunct synbiotics improved clinical and immunological outcomes in gingivitis patients, irrespective of smoking status.
Grusovin, 2019 [64] Italy	Periodontitis	10	10	*L. reuteri*	3 months, 3-month washout, 3 months/ 3 months, 9 months	Lozenge	FM-GBT	Yes	BOP ^+^, PPD ^+^, PAL ^+^, tooth survival	Probiotics improved clinical parameters with periodontal maintenance therapy.
Hallström, 2013 [42] Sweden	Healthy	9	9	*L. reuteri*	3 weeks/ 3 weeks	Lozenge	None	No	PI, GI, BOP, GCF, IL-1β *, IL6, IL8 *, IL10, IL-18 *, MIP-1β *, TNF-α, *A. actinomycetemcomitan*, *A. naeslundii*, *C. rectus*, *F. alocis*, *F. nucleatum **, *L. reuteri*, *L. fermentum*, *P. micra*, *P. endodontis*, *P. intermedia*, *P. gingivalis*, *S. intermedia*, *S. mutans*, *S. oralis **, *S. sanguinis*, *T. forsythia*, *T. denticola*, *V. parvula **	Probiotic supplementation did not significantly affect plaque accumulation, inflammatory reactions in the gingiva, and the microbiological composition in healthy individuals with experimental gingivitis.
Ikram, 2018 [65] Pakistan	Periodontitis	15	15	*L. reuteri*	3 months/ 4 months	Sachet	SRP + amoxicillin + metronidazole	Yes	PlI *, CAL *, BOP *, PPD *	Probiotics showed similar efficacy in the improvement of periodontal clinical outcomes as antibiotics.
Ikram, 2019 [66] Pakistan	Periodontitis	14	14	*L. reuteri*	12 weeks/ 12 weeks	Sachet	SRP	Yes	PlI *, CAL ^+^, BOP ^+^, PPD ^+^	Probiotics may be used as an adjunctive treatment with SRP to treat chronic periodontitis.
Ince, 2015 [67] Turkey	Periodontitis	15	15	*L. reuteri*	3 weeks/ 3 weeks	Lozenge	SRP	Yes	PlI ^+^, GI ^+^, BOP ^+^, PPD ^+^, CAL ^+^, GCF *, MMP-8 ^+^, TIMP-1 ^+^	Adjuvant probiotic treatment improved clinical and immunological outcomes in periodontitis patients.
Iniesta, 2012 [43] Spain	Gingivitis	20	20	*L. reuteri*	4 weeks/ 4 weeks	Chewing tablet	None	No	PlI, GI, *Lactobacillus* spp., *A. actinomycetemcomitan*, *C. rectus*, *Capnocytophaga* spp., *E. corrodens*, *F. nucleatum*, *P. micra*, *P. intermedia*, *P. gingivalis*, *Tannerella forsythia*, total bacteria	Probiotic administration reduced subgingival periodontopathogen count.
Invernici, 2018 [68] Brazil	Periodontitis	20	21	*B. lactis*	30 days/ 30 days	Lozenge	SRP	Yes	PlI ^+^, CAL ^+^, PPD ^+^, BOP ^+^, REC, IL-1β ^+^, IL-8 ^+^, IL-10 *, *B. animalis ^+^*,	Probiotic supplementation in addition to SRP may improve clinical, microbiological, and immunological outcomes in generalized chronic periodontitis patients.
Iwasakia, 2016 [69] Japan	Periodontitis	19	17	*L. plantarum*	12 weeks/ 12 weeks	Capsule	SPT	Not mentioned	PlI, GI, BOP, PPD ^+^	Chronic periodontitis patients with adjunctive probiotic treatment may lead to improvements in periodontal pockets.
Jagadeesh, 2017 [70] India	Gingivitis	15	15	*B. coagulans*	3 weeks/ 3 weeks	Chewing tablet	None	Not mentioned	PlI, GI *, BOP ^+^, GPx	Probiotic use led to a statistically significant decrease in BOP.
Jäsberg, 2018 [71] Finland	Healthy	29	31	*B. animalis*, *L. rhamnosus*	4 weeks/ 4 weeks	Lozenge	None	Not mentioned	PlI ^+^, GI ^+^, MMP-8, MMP-9 ^+^, TIMP-1 ^+^, *S. mutans*, *lactobacilli*	Probiotics may immunomodulate the oral cavity.
Keller, 2018 [72] Denmark	Gingivitis	23	24	*L. curvatus*, *L. rhamnosus*	4 weeks/ 4 weeks	Tablet	None	No	PlI *, BOP ^+^, GCF ^+^, IL-1β, IL-6, IL-8, IL-10, TNF-α	Probiotic use may improve gingival health without affecting the oral microbiome and immune response.
Krasse, 2005 [73] Sweden	Gingivitis	20	18	*L. reuteri*	14 days/ 14 days	Chewing gum	None	Yes	PlI ^+^, GI ^+^, *L. reuteri ^+^*, Total *lactobacillus* ^+^	*L. reuteri* can reduce PlI and GI in gingivitis patients.
Kuka, 2019 [74] Turkey	Periodontitis	18	18	*L. reuteri*	3 weeks/ 12 weeks	Tablet	IPT	Yes	BOP ^+^, PPD ^+^, GCF ^+^, NO ^+^	Probiotics may be an adjunct to IPT. NO in GCF is a potential inflammatory marker in periodontal diseases.
Kuru, 2017 [75] Turkey	Healthy	26	25	*B. animalis*	4 weeks/ 4 weeks	Yogurt	None	Yes	PlI ^+^, GI ^+^, BOP ^+^, PPD ^+^, GCF ^+^, IL-1β ^+^	Probiotics improved clinical and immunological outcomes compared to controls after a 5-day non-brushing period.
Laleman, 2015 [76] Turkey	Periodontitis	24	24	*S. oralis*, *S. rattus*, *S. uberis*	12 weeks/ 12 weeks	Tablet	SRP	Not mentioned	CAL *, BOP *, PPD *, REC *, *F. nucleatum* *, *P. gingivalis* *, *P. intermedia ^+^*, *T. forsythia* *	Probiotic formulation used did not show statistically significant improvements in clinical or microbiological outcomes.
Laleman, 2019 [77] Belgium	Periodontitis	19	20	*L. reuteri*	12 weeks/12 weeks	Lozenge	NSPT	Yes	PlI *, CAL *, BOP *, PPD ^+^, REC *, *A. actinomycetemcomitans*, *F. nucleatum*, *P. intermedia*, *P. gingivalis*	Adjunctive use of probiotics after NSPT reduced PPD and the percentage of sites in need of surgery.
Lee, 2015 [78] Korea	Healthy	14	16	*L. brevis*	14 days/ 14 days	Lozenge	Scaling and polishing	Yes	PlI *, GI *, BOP *, NO, MMP-8, PGE2 *	Probiotic supplementation may decrease inflammatory cascades through NO and PGE2.
Mayanagi, 2009 [79] Japan	Periodontitis	34	32	*L. salivarius*	8 weeks/ 8 weeks	Tablet	None	No	*A. actinomycetemcomitans*, *P. intermedia*, *P. gingivalis*, *T. forsythia ^+^*, *T. denticola*, total bacteria *	Probiotics decreased the subgingival *T. forsythia* count at 4 and 8 weeks and the total bacteria count at 4 weeks.
Meenakshi, 2018 [80] India	Periodontitis	10	10	*L. casei*	1 month/ 1 month	Drink	SRP	No	PlI ^+^, GI ^+^, CAL ^+^, PPD^+^, total bacteria^+^	Probiotics as an adjunct to SRP improved clinical outcomes and reduced total bacterial count.
Mitic, 2017 [81] Macedonia	Periodontitis	15	15	*B. bifidum*, *B. coagulans*, *L. acidophilus*, *L. bulgaricus*, *S. thermophilus*	15 days/ 1 month	Lozenge	SRP	Yes	PlI *, GI *, GBI *, CAL *, PPD ^+^, anaerobic bacterial count^+^	Probiotics may improve clinical outcomes and bacterial count in periodontitis patients.
Montero, 2017 [14] Spain	Gingivitis	30	29	*L. brevis*, *L. plantarum*, *P. acidilactici*	6 weeks/ 6 weeks	Chewing tablet	PMPR	Yes	PlI *, GI *, AngBs ^+^, *A. actinomycetemcomitans **, *C. rectus*, *Fusobacterium* spp., *P. gingivalis*, *T. forsythia **	Decreased number of sites with severe inflammation compared to placebo group in gingivitis patients. Decreased *T. forsythia* count.
Morales, 2016 [82] Chile	Periodontitis	14	14	*L. rhamnosus*	3 months/ 3 months	Sachet	SRP	Yes	CAL *, PlI *, BOP, PPD *	Probiotic use improved clinical symptoms similar to SRP alone.
Morales, 2017 [10] Chile	Periodontitis	16	15	*L. rhamnosus*	3 months/ 9 months	Sachet	1) SRP 2) SRP + Antibiotic	Yes	CAL *, BOP, PPD *, PA *, *A. actinomycetemcomitans*, *P. gingivalis* *, *T. forsythia*, total bacteria *	Probiotic and antibiotic groups had similar clinical and microbiological improvements to placebo.
Nadkerny, 2015 [83] India	Gingivitis	15	15	*B. longum*, *L. acidophilus*, *L. rhamnosus*, *L. sporogenes*, *S. boulardii*	4 weeks/ 4 weeks	Sachet	Scaling and polishing 1) Chlorhexidine 2) Normal saline	Yes	PlI ^+^, GI ^+^, OHI-S ^+^	Probiotic mouthwash effectively reduced plaque accumulation and gingival inflammation.
Nasry, 2018 [84] Egypt	Gingivitis	20	20	*L. rhamnosus*	2 weeks/ 2 weeks	Spray	Scaling and polishing	Yes	PlI ^+^, GI ^+^, SI ^+^	Miswak and probiotic formulation led to the greatest reduction in plaque and gingival indices.
Pelekos, 2019 [34] Hong Kong	Periodontitis	21	20	*L. reuteri*	28 days/ 90 days	Lozenge	NSPT	Yes	CAL *, BOP *, PPD *	Adjunctive use of probiotics did not show increased effectiveness compared to control.
Pelekos, 2020 [85] Hong Kong	Periodontitis	20	20	*L. reuteri*	28 days/ 90 days	Lozenge	NSPT	Yes	CAL^+^, BOP *, PPD ^+^	Probiotic supplementation improved periodontal pockets ≥ 5 mm and CAL.
Penala, 2015 [86] India	Periodontitis	15	14	*L. reuteri*, *L. salivarius*	15 days/ 3 months	Capsule & Mouthwash	SRP	Yes	PlI ^+^, MGI ^+^, GBI ^+^, PPD *, CAL *, BANA, ORG	Probiotic use improved clinical outcomes and oral malodor parameters.
Pudgar, 2020 [87] Slovenia	Periodontitis	20	20	*L. brevis*, *L. plantarum*	Once (gel) 3 months (lozenge)/ 3 months	Local gel & Lozenge	SRP	Yes	DS *, PlI *, CAL *, BOP *, PPD *, REC*, GBI *	Probiotic and control groups both had significant clinical improvements, but there was no statistically significant difference between the two groups.
Sabatini, 2017 [88] Italy	Gingivitis	40	40	*L. reuteri*	30 days/ 30 days	Tablet	None	Yes	PlI ^+^, BOP ^+^	Probiotics were effective in reducing plaque and BOP in type 2 diabetes patients with gingivitis.
Sajedinejad, 2017 [89] Iran	Periodontitis	10	10	*L. salivarius*	4 weeks/ 4 weeks	Mouthwash	SRP	Yes	PlI, GI ^+^, BOP ^+^, PPD *, *A. actinomycetemcomitans ^+^*	Probiotic use improved clinical and microbiological outcomes.
Scariya, 2015 [90] India	Gingivitis and Periodontitis	14	14	*S. salivarius*	30 days/ 30 days	Tablet	None	Yes	PlI ^+^, GI ^+^, SBI ^+^, PPD^+^	Probiotic use improved clinical outcomes compared to controls.
Schlagenhauf, 2018 [91] Germany	Gingivitis	24	21	*L. reuteri*	Within 2 days after delivery (41.9 ± 16.0 days)	Lozenge	None	No	PlI ^+^, GI ^+^, TNF-α	Probiotics may be a useful adjunct for pregnancy-related gingivitis.
Schlagenhauf, 2020 [92] Germany	Gingivitis & Periodontitis	33	35	*L. reuteri*	42 days/ 42 days	Lozenge	None	No	PCR ^+^, GI ^+^, BOP ^+^, PAL ^+^, PPD ^+^	Probiotic use improved all clinical outcomes compared to controls.
Shah, 2013 [93] India	Periodontitis	10	10 (Control) 10 (Antibiotic)	*L. brevis*	2 weeks/ 2 months	Tablet	SRP 1) Probiotic + Doxycycline 2) Doxycycline alone	No	PlI *, GI *, CAL *, PPD *, *lactobacilli^+^*, *A. actinomycetemcomitans* *	Probiotic use decreased clinical and microbiological parameters when used alone or in combination with doxycycline.
Shah, 2017 [94] India	Periodontitis	6	6	*L. brevis*	14 days/ 5 months	Lozenge	SRP 1) Probiotics + Doxycycline 2) Doxycycline alone	No	GI ^+^, PlI, PPD, CAL, *A. actinomycetemcomitans*, *Lactobacillus* spp.	No synergy at 5 months when probiotics and doxycycline were both given. No statistically significant difference between antibiotic and probiotic supplementation.
Shetty, 2020 [95] India	Periodontitis	60	60	*B. mesentericus*, *C. butyricum*, *L. sporogenes. S. faecalis*	Once (local)/ 3 months	Local	SRP	Not mentioned	PlI *, GI *, PPD *, IL-6 ^+^, ALP *, *P.Gingivalis* *, *P. intermedia* *	Synbiotic treatment may improve clinical, microbiological, and immunological outcomes in patients with chronic periodontitis.
Shimauchi, 2008 [96] Japan	Healthy	34	32	*L. salivarius*	8 weeks/ 8 weeks	Tablet	None	No	PlI *, GI *, BOP *, PPD *, *L. salivarius ^+^*, Lactoferritin * (Saliva)	Probiotics may be useful for maintenance and/or improvement of oral health in individuals at risk of periodontal diseases.
Sinkiewicz, 2010 [97] Sweden	Healthy	11	12	*L. reuteri*	12 weeks/ 12 weeks	Chewing gum	None	No	PlI, *A. naeslundii* *, *A. actinomycetemcomitans* *, *C. rectus* *, *F. alocis* *, *F. nucleatum* *, *L. acidophilus*, *L. fermentum **, *L. reuteri **, *P. micra*, *P. gingivalis* *, *P. endodontalis* *, *P. intermedia* *, *T. forsythia* *, *T. denticola* *, *S. intermedia*, *S. mutans **, *S. oralis*, *S. sanguinis* *, *V. parvula* ***	There was a statistically significant increase in plaque in the controls, but not the probiotics group. No changes between probiotics and control groups in the oral microbiota.
Slawik, 2011 [98] Germany	Healthy	11	17	*L. casei*	14 days/ 14 days	Drink	None	No	PlI *, GI *, BOP ^+^, GCF^+^	Probiotics may have an anti-inflammatory effect.
Snulingga, 2020 [99] Indonesia	Periodontitis	8	8	*L. reuteri*	14 days/ 14 days	Lozenge	SRP	Not mentioned	CAL ^+^, IL-4	Probiotic use as an adjunct decreased CAL and increased IL-4.
Staab, 2009 [100] Germany	Healthy	25	25	*L. casei*	8 weeks/ 8 weeks	Drink	None	No	PlI ^+^, PBI *, MPO ^+^, MMP-3 ^+^, Elastase *	Probiotics may improve periodontal health through immunomodulation.
Suzuki, 2012 [101] Japan	Periodontitis	20	22	*L. salivarius*	2 weeks/ 2 weeks	Oil drops	None	No	BOP ^+^, PPD *, *Ubiquitous bacteria* *, *F. nucleatum*, *P. gingivalis*, *L. salivarius **, *P. intermedia*, *S. mutans*, *T. forsythia*, *T. denticola*	Probiotics improved BOP and had a decreased periodontopathogen count compared to controls.
Tekce, 2015 [41] Turkey	Periodontitis	20	20	*L. reuteri*	3 weeks/ 3 weeks	Lozenge	SRP	Yes	PlI ^+^, GI ^+^, BOP ^+^, PPD ^+^, RAL *, Anaerobic bacteria ^+^, TVC ^+^	Probiotics as an adjuvant can improve clinical and microbiological outcomes.
Teughels, 2013 [102] Turkey	Periodontitis	15	15	*L. reuteri*	12 weeks/ 12 weeks	Lozenge	SRP	Yes	PlI *, CAL ^+^, GBI ^+^, BOP *, PPD^+^, REC, *A. actinomycetemcomitans **, *F. nucleatum **, *T. forsythia **, *P. gingivalis ^+^*, *P. intermedia **, Total bacteria *	Probiotics as an adjuvant can improve clinical and microbiological outcomes.
Theodoro, 2019 [103] Brazil	Periodontitis	14	14	*L. reuteri*	21 days/ 90 days	Chewing tablet	SRP	Yes	BOP ^+^, CAL, PPD ^+^, REC	Adjuvant use of probiotics to treat chronic periodontitis in smokers reduced gingival inflammation.
Tobita, 2018 [104] Japan	Healthy	8	8	*L. crispatus*	4 weeks/ 4 weeks	Tablet	None	No	PS ^+^, *A. actinomycetemcomitans*, *F. nucleatum **, *T. forsythia*, *P. gingivalis ^+^*, *P. intermedia*, *T. denticola*.	Probiotic use can improve the oral environment and hence may help prevent periodontal disease.
Toiviainen, 2015 [105] Finland	Healthy	29	31	*B. lactis, L. rhamnosus*	4 weeks/ 4 weeks	Lozenge	None	Not mentioned	PlI^*^, GI^*^, *Lactobacillus, S. mutans*	Probiotics improved clinical outcomes but not microbiological.
Twetman, 2009 [106] Denmark	Gingivitis	14	13	*L. reuteri*	2 weeks/ 2 weeks	Chewing gum	None	Yes	BOP^*^, IL-1β, TNF-α, GCF^*^, IL-6^*^, IL-8^*^, IL-10	Probiotics are beneficial to gingival health in a dose dependent manner.
Vicario, 2013 [107] Spain	Periodontitis	10	9	*L. reuteri*	1 month/ 1 month	Tablet	None	Yes	PlI^*^, BOP^*^, PPD^*^	Probiotic supplementation can improve inflammatory and clinical outcomes in patients with mild to moderate periodontitis.
Vivekananda, 2010 [108] India	Periodontitis	15	15	*L. reuteri*	21 days/ 42 days	Lozenge	1) SRP 2) Without SRP	Yes	PlI^*^, GI^*^, GBI^*^, CAL^*^, PPD^*^, *A. actinomycetemcomitans*^*^*, P. gingivalis*^*^*, P. intermedia*^*^	Probiotic use can improve periodontal health through plaque inhibition, anti-inflammatory and antimicrobial effects.
Vohra, 2019 Saudi [109] Arabia	Periodontitis	31	32	*L. reuteri*	21 days/ 3 months	Lozenge	SRP	Yes	PlI^*^, CAL^*^, BOP^*^, PPD^*^	Probiotic use is not an effective adjunct to SRP in chronic periodontitis patients.
Yuki, 2019 [110] Japan	Periodontal disease	12	11	*L. rhamnosus*	90 days/ 90 days	Yogurt	None	Yes	GI^*^, PPD^*^, PMA^+^	Probiotic use improved clinical parameters under study.

Note: * Indicates a statistically significant difference within the probiotic group from baseline to follow-up. + Indicates a statistically significant difference between the probiotic and control groups. Abbreviations: Clinical—AngBs, Angulated bleeding score; BOP, Bleeding on probing; CAL, Clinical attachment level; DS, Disease sites defined as probing pocket depth > 4 mm and BOP; GBI, Gingival bleeding index; GCF, Gingival crevicular fluid; GI, Gingival index; MGI, Modified gingival index; OHI, Oral hygiene index; OHI-S, Oral hygiene index simplified; PAL, Probing attachment level; PCR, Plaque control record; PI, Periodontal index; PA, Plaque accumulation; PBI, Papillary bleeding index; PlI, Plaque index; PMA, Papillary-marginal-attached index; PPD, Probing pocket depth; PS, Plaque score; RAL, Relative attachment level; REC, Gingival recession; SBI, Sulcular bleeding index; SI, Stain index; Microbiological—*A. actinomycetemcomitans*, *Aggregatibacter actinomycetemcomitans*; *A. naeslundii*, *Actinomyces naeslundii*; *B. subtilis*, *Bacillus subtilis*; *B. megaterium*, *Bacillus megaterium*; *B. mesentericus*, *Bacillus mesentericus*; *B. pumulus*, *Bacillus pumulus*; *B. animalis*, *Bifidobacterium animalis*; *B. bifidum*, *Bifidobacterium bifidum*; *B. coagulans*, *Bacillus coagulans*; *B. lactis*, *Bifidobacterium lactis*; *B. longum*, *Bifidobacterium longum*; *C. rectus*, *Campylobacter rectus*; *C. butyricum*, *Clostridium butyricum*; *E. corrodens*, *Eikenella corrodens*; *E. faecium*, *Enterococcus faecium*; *F. alocis*, *Filifactor alocis*; *F. nucleatum*, *Fusobacterium nucleatum*; *L. acidophilus*, *Lactobacillus acidophilus*; *L. brevis*, *Lactobacillus brevis*; *L. bulgaricus*, *Lactobacillus bulgaricus*; *L. casei*, *Lactobacillus casei*; *L. crispatus*, *Lactobacillus crispatus*; *L. curvatus*, *Lactobacillus curvatus*; *L. delbrueckii*, *Lactobacillus delbrueckii*; *L. fermentum*, *Lactobacillus fermentum*; *L. plantarum*, *Lactobacillus plantarum*; *L. rhamnosus*, *Lactobacillus rhamnosus*; *L. reuteri*, *Lactobacillus reuteri*; *L. salivarius*, *Lactobacillus salivarius*; *L. sporogenes*, *Lactobacillus sporogenes*; *L. lactis*, *Lactococcus lactis*; *P. acidilactici*, *Pediococcus acidilactici*; *P. endodontalis*, *Porphyromonas endodontalis*; *P. gingivalis*, *Porphyromonas gingivalis*; *P. intermedia*, *Prevotella intermedia*; *P. micra*, *Parvimonas micra*; *S. faecalis*, *Streptococcus faecalis*; *S. intermedia*, *Streptococcus intermedia*; *S. mutans*, *Streptococcus mutans*; *S. oralis*, *Streptococcus oralis*; *S. rattus*, *Streptococcus rattus*; *S. salivarius*, *Streptococcus salivarius*; *S. sanguinis*, *Streptococcus sanguinis*; *S. thermophilus*, *Streptococcus thermophilus*; *S. uberis*, *Streptococcus uberis*; *S. boulardii*, *Saccharomyces boulardii*; *T. forsythia*, *Tannerella forsythia*; *T. denticola*, *Treponema denticola*; *V. parvula*, *Veillonella parvula*; Immunological—CAT, Catalase; GPx, Glutathione peroxidase; IL, Interleukin; MDA, Malondialdehyde; MIP-1β, Macrophage inflammatory protein 1 beta; MMP-8, matrix metalloproteinase-8; MPO, myeloperoxidase; NO, Nitric oxide; SOD, Super-oxide dismutase; TAC, Total antioxidant capacity; TIMP-1, Tissue inhibitor of metalloproteinase; TNF-α, Tumor necrosis factor alpha; Other—BANA, N-benzoyl-DL-arginine-naphthylamide; ORG, Halitosis assessment with organoleptic scores; FM-GBT, Full mouth guided biofilm therapy; IPT, Initial periodontal therapy; NSPT, Non-surgical periodontal therapy; PMPR, Professional manual plaque removal; RSD, Root surface debridement; SPT, Supporting periodontal therapy; SRP, Scaling and root planing; TVC, Total viable count.

**Table 2 nutrients-14-01036-t002:** Subgroup analysis examining the effects of probiotic supplementation on clinical outcomes.

Clinical Outcomes	Subgroup	Level of Subgroup	SMD	95% CI	I^2^	*p*-Value	Sample Size	
							Probiotic	Control
Plaque index (PlI)	Type of periodontal disease	Gingivitis	0.153	−0.152, 0.457	20.906	0.281	108	99
		Periodontitis	0.736	0.267, 1.206	71.842	**0.001**	136	135
	Type of probiotic strain	*Lactobacillus*	0.639	0.169, 1.110	75.533	**<0.001**	154	151
		Mixed	0.280	−0.159, 0.719	0.000	0.523	42	36
		Other	0.185	−0.212, 0.582	0.000	0.431	48	47
	Type of *Lactobacillus* species	*L. Reuteri*	0.707	0.034, 1.381	80.976	**<0.001**	98	95
		Other	0.590	−0.456, 1.636	81.557	**0.004**	42	42
	Treatment duration	≤1 month	0.615	0.146, 1.084	75.448	**<0.001**	154	153
		>1 to 2 months	0.328	−0.006, 0.661	0.000	0.406	73	64
		> 2 months	0.053	−0.603, 0.710	0.000	1.000	17	17
	Mode of delivery	Ingestion	0.952	−0.894, 2.797	89.037	**0.003**	27	27
		Local	0.323	−0.202, 0.847	0.000	1.000	28	27
		Oral	0.239	−0.302, 0.780	0.251	0.251	34	34
		Oral and Ingestion	0.495	0.061, 0.930	0.001	**<0.001**	155	146
	Oral hygiene instructions	Yes	0.622	0.204, 1.040	66.923	**0.006**	145	138
		No	0.665	−0.415, 1.746	85.436	**<0.001**	54	51
Mean plaque percentage (MPP)	Type of periodontal disease	Gingivitis	1.279	−0.905, 3.463	96.629	**<0.001**	63	64
		Periodontitis	0.681	0.072, 1.290	82.212	**<0.001**	130	130
	Type of probiotic strain	*Lactobacillus*	1.037	0.391, 1.683	88.278	**<0.001**	195	198
		Mixed	0.112	−0.728, 0.952	0.000	1.000	10	10
		Other	0.199	−0.396, 0.794	0.000	1.000	21	21
	Type of *Lactobacillus* species	*L. Reuteri*	1.458	0.724, 2.191	86.723	**<0.001**	148	150
		Other	0.193	−0.200, 0.586	0.000	0.889	47	48
	Treatment duration	≤1 month	0.937	0.076, 1.798	90.960	**<0.001**	145	146
		>1 to 2 months	1.560	1.022, 2.099	0.000	1.000	33	35
		> 2 months	0.460	0.008, 0.912	21.718	0.279	48	48
	Mode of delivery	Ingestion	0.969	−0.506, 2.445	94.211	**<0.001**	77	78
		Oral	0.537	−0.348, 1.423	59.024	0.118	24	24
		Oral and Ingestion	0.942	0.159, 1.725	87.844	**<0.001**	125	127
	Oral hygiene instructions	Yes	0.880	0.197, 1.564	88.210	**<0.001**	170	170
		No	0.865	−0.502, 2.232	91.878	**<0.001**	56	59
Gingival index (GI)	Type of periodontal disease	Gingivitis	0.298	−0.089, 0.684	49.985	0.092	108	99
		Periodontitis	1.069	0.296, 1.841	86.299	**<0.001**	116	112
	Type of probiotic strain	*Lactobacillus*	1.236	0.574, 1.897	87.366	**<0.001**	178	174
		Mixed	0.101	−0.333, 0.535	0.000	0.949	43	36
		Other	0.329	−0.070, 0.729	0.000	0.354	48	47
	Type of *Lactobacillus* species	*L. Reuteri*	1.621	0.648, 2.595	89.871	**<0.001**	112	111
		Other	0.817	0.018, 1.616	79.137	**0.001**	66	63
	Treatment duration	≤1 month	0.949	0.270, 1.628	85.079	**<0.001**	132	130
		>1 to 2 months	0.900	−0.116, 1.915	91.498	**<0.001**	106	99
		>2 months	0.888	−0.920, 2.696	89.949	**0.002**	31	28
	Mode of delivery	Ingestion	1.258	−0.169, 2.686	87.547	**<0.001**	41	38
		Local	0.494	−0.035, 1.023	0.000	1.000	28	27
		Oral	0.189	−0.305, 0.682	0.000	0.674	30	30
		Oral and Ingestion	1.051	0.306, 1.797	89.846	**<0.001**	170	162
	Oral hygiene instructions	Yes	1.051	0.327, 1.775	86.466	**<0.001**	134	126
		No	1.344	0.261, 2.427	89.898	**<0.001**	87	86
Pocket probing depth (PPD)	Type of periodontal disease	Gingivitis	0.997	−0.853, 2.848	92.406	**<0.001**	35	38
		Periodontitis	0.578	0.355, 0.801	62.720	**<0.001**	442	434
	Type of probiotic strain	*Lactobacillus*	0.674	0.386, 0.962	69.524	**<0.001**	330	329
		Mixed	0.387	0.045, 0.729	0.000	0.740	67	62
		Other	0.379	−0.037, 0.795	51.535	0.103	92	92
	Type of *Lactobacillus* species	*L. Reuteri*	0.677	0.315, 1.040	74.541	**<0.001**	249	252
		Other	0.657	0.169, 1.144	56.911	**0.041**	81	77
	Treatment duration	≤1 month	0.737	0.430, 1.044	66.736	**<0.001**	270	264
		>1 to 2 months	0.514	−0.030, 1.059	65.160	0.057	76	79
		> 2 months	0.326	0.015, 0.636	43.082	0.080	143	140
	Mode of delivery	Ingestion	0.514	0.106, 0.922	47.870	0.088	94	91
		Local	0.919	0.370, 1.468	0.000	1.000	28	27
		Oral	0.918	−0.071, 1.907	79.291	**0.008**	44	44
		Oral and Ingestion	0.525	0.251, 0.800	66.577	**<0.001**	323	321
	Oral hygiene instructions	Yes	0.592	0.343, 0.841	63.935	**<0.001**	366	360
		No	0.953	0.308, 1.597	65.707	0.054	66	69
	Disease severity	Deep	0.735	0.209, 1.261	73.585	**0.002**	112	114
		Moderate	0.499	0.043, 0.955	66.202	**0.011**	112	114
Clinical attachment level (CAL)	Type of probiotic strain	*Lactobacillus*	0.417	0.225, 0.609	8.881	0.355	229	228
		Mixed	0.395	−0.066, 0.855	0.000	0.401	38	34
		Other	0.415	0.076, 0.755	7.610	0.339	72	72
	Type of *Lactobacillus* species	*L. Reuteri*	0.416	0.201, 0.631	12.027	0.330	189	189
		Other	0.445	−0.086, 0.975	31.016	0.235	40	39
	Treatment duration	≤1 month	0.388	0.185, 0.592	0.000	0.547	186	181
		>1 to 2 months	0.789	0.236, 1.343	34.507	0.217	41	41
		> 2 months	0.330	0.071, 0.588	0.000	0.571	112	112
	Mode of delivery	Ingestion	0.464	0.116, 0.812	0.276	0.390	63	63
		Local	0.696	0.159, 1.233	0.000	1.000	28	27
		Oral	0.887	0.132, 1.643	0.000	1.000	14	14
		Oral and Ingestion	0.339	0.159, 0.520	0.000	0.543	234	230
	Oral hygiene instructions	Yes	0.351	0.178, 0.523	0.000	0.789	256	251
		No	0.835	0.437, 1.233	0.000	0.376	51	51
	Disease severity	Deep	0.373	0.088, 0.657	0.000	0.690	92	94
		Moderate	0.422	0.137, 0.706	0.000	0.886	92	94
Bleeding on probing (BOP)	Type of periodontal disease	Gingivitis	0.685	−0.438, 1.808	93.899	**<0.001**	117	120
		Periodontitis	0.749	0.404, 1.094	72.526	**<0.001**	260	257
	Type of probiotic strain	*Lactobacillus*	0.878	0.442, 1.313	85.057	**<0.001**	314	312
		Mixed	0.035	−0.574, 0.643	0.000	1.000	19	21
		Other	0.202	−0.210, 0.613	0.000	0.640	44	44
	Type of *Lactobacillus* species	*L. Reuteri*	1.054	0.485, 1.622	86.818	**<0.001**	217	217
		Other	0.502	−0.078, 1.081	74.262	**0.002**	97	95
	Treatment duration	≤1 month	1.024	0.454, 1.595	88.021	**<0.001**	236	238
		>1 to 2 months	0.095	−0.513, 0.703	0.000	1.000	20	20
		> 2 months	0.402	0.020, 0.785	55.314	**0.037**	121	119
	Mode of delivery	Ingestion	0.742	−0.391, 1.876	93.499	**<0.001**	112	110
		Oral	1.166	−0.037, 2.370	89.525	**<0.001**	63	65
		Oral and Ingestion	0.616	0.296, 0.936	60.339	**0.005**	202	202
	Oral hygiene instructions	No	0.054	−0.508, 0.617	0.000	1.000	23	24
		Yes	0.966	0.478, 1.454	86.250	**<0.001**	277	276
Gingival crevicular fluid (GCF)	Type of periodontal disease	Gingivitis	0.626	0.162, 1.091	0.000	0.392	36	36
		Periodontitis	0.507	0.027, 0.986	0.000	0.496	33	33

Subgroup analysis assessing the effects of probiotic supplementation compared to a control on clinical outcomes in periodontal diseases using a random-effects model based on the: 1. Type of periodontal disease; 2. Type of probiotic strain; 3. Type of *Lactobacillus* species; 4. Treatment duration; 5. Mode of probiotic delivery; and 6. Oral hygiene instruction. Note: Bold indicates statistically significant findings (*p*-value ≤ 0.05). SMD, Hedge’s g standardized mean difference. I^2^, Measure of heterogeneity.

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
