# Peer review of "The Clinical, Microbiological, and Immunological Effects of Probiotic Supplementation on Prevention and Treatment of Periodontal Diseases: A Systematic Review and Meta-Analysis"

_nutrients, 2022, doi:10.3390/nu14051036_

Round 1

Reviewer 1 Report

nutrients-1573684-peer-review-v1

The manuscript “The clinical, microbiological and immunological effects of probiotic treatment on prevention and treatment of periodontal diseases: A comprehensive systematic review and meta-analysis” evaluated the effects of probiotic supplementation in the prevention and treatment of periodontal diseases based on the assessment of clinical, microbiological and immunological outcomes.

Overall, this work needs major improvements. A major concern is the length of the manuscript. Overall it is almost 40 pages of partially non structured text. The authors failed to focus on the main content in the Introduction, M&M, Results and Discussion section. It is hard to follow/understand the pure mass of findings and almost impossible to critically evaluate the work. The authors are strongly encouraged to tailor down on specific questions that have not yet been reviewed in the literature. It seems to the reviewer (see supplement) that the authors wanted to give a full review of everything that concerns probiotics and periodontal disease. However, this is beyond the scope of a single manuscript, especially when the data are presented in way that makes it hard to follow the train of thought.       

Titel:

  • “The clinical, microbiological and immunological effects of probiotic treatment on prevention and treatment of periodontal diseases: A comprehensive systematic review and meta-analysis.” Please change ‘treatment’ to ‘supplementation’.

Abstract:

  • line 32/33: Bacteria are written in italics

Introduction:

  • Line 47: “…as the global population ages[8,9].” Space is missing.
  • Line 71-73: “During the progression of the periodontal disease, subgingival bacterial counts could increase from 1x103 in a healthy gum crevice, up to 1x108 in a periodontal pocket [13].” Please cite the original reference (Socransky SS, Haffajee AD. Microbiology of periodontal disease. In: Lindhe J, Karring T, Lang NP, eds. Clinical Periodontology and Implant Dentistry. Copenhagen, Denmark: Munksgaard Blackwells, 2003). Please check if the numbers are really stated in the original publication. Also this work is from 2003 – there have been tremendous changes in oral biofilm research and it is proven that biofilm diversity decreases in periodontal diseased patients. Are you referring solely to gram-negative and anaerobe subgingival bacterial counts?
  • Please use subheadings in the introduction – this way it is way more easy to follow the introduction.
  • Line 45-73: Please shorten this section
  • Line 74-94: Bacteria are written in italics. Please check the full MS.
  • Line 74-76: “The primary etiology of periodontal disease is an imbalanced oral microbiome population developing progressively over time with an increasing number of gram negative anaerobic bacteria that disrupts the microbiota-host homeostasis [19,20].” Please be more precisely: It is not the whole oral microbiome, but rather the subgingival microbiome which exhibits an imbalance. The authors describe the process, which leads to a dysbiotic biofilm – please briefly explain what is happening to healthy bacteria or non-periopathogens.
  • Line 85-89: “Furthermore, dental calculus which is composed of calcified dental plaque surrounded by a nonmineralized bacterial layer, is a predisposing factor for periodontal disease [22]. It is formed through deposition of calcium phosphate mineral salts from saliva and gingival fluid on pre-existing dental plaque, facilitating bacterial involvement through its porous structure and the development of periodontal disease [22].” The authors should consider to re-locate this section either at the very beginning of the second paragraph or fully cancel it.
  • Line 90-95: “The imbalanced inflammatory response in periodontal disease is likely to underly the reciprocal link or crosstalk with other chronic systemic diseases [23]. Periodontal diseases share common inflammatory risk factors with various chronic systemic conditions, especially those with impaired immune function including diabetes, rheumatoid arthritis, cardiovascular disease, osteoporosis, and HIV-infection [13,24].” The authors should re-locate this paragraph to the first section, where they already explained the connection between oral and systemic health. Also please check if the imbalanced inflammatory response is the only link between those entities, because literature also discusses certain pathogens and their virulence factors. In general, please organize your introduction better and use subheadings. It is difficult to follow your introduction, because you often jump between different topics.
  • Line 95-115: This sections describes periodontal classification. (1) Please re-locate second paragraph (107-115) before Staging and Grading. (2) Line 97: explanations of all abbreviations are missing? (3) Are the authors describing the current gold-standard for periodontal classification? Why did they choose this classification system? Why are the only describing Staging and not Grading? Please shorten the whole paragraph for periodontal classification!
  • Line 116-128: This sections describes the periodontal therapy – however I would strongly advise to either discuss this part in the discussion (when necessary) or delete it completely. The current Introduction is way too long and distracts strongly from the actual content of the work, which is the clinical, microbiological and immunological effects of probiotic treatment. The immunological effects are not presented in the introduction!
  • Line 130-148: At the very end of a very long introduction the authors introduce the use of probiotic treatment in periodontal disease. It is well written and easy to understand – however what are the clinical and immunological benefits? This is part of your title and is not addressed in the Introduction?

Material &Methods:

  • Line 196-200: Bacteria are written in italics.
  • Line 218-226: I would either re-locate this paragraph to the supplements or delete it. The M&M section is again too long and internal calibration is not necessary to understand the researchers work.
  • Line 230: Microsoft Excel Version 2102?
  • Line 234-236: Bacteria are written in italics.
  • Line 245-280: Please shorten statistical strategy.
  • Line 294-316: Please shorten additional analysis.

Results:

  • Line 381: The authors divided the presentation of results according to clinical outcome, microbiological outcome and immunological outcome, which makes is easy for the reader to follow your findings. However, please focus just on the main finding: e.g. Probiotic use and Plaque Index: Only state 1.A. Please put the rest in supplementals or maybe reconsider the importance of the subgroup analysis. Only the results for clinical parameters is more than 12 pages long!!! This is just too much information in one manuscript. This mass of results makes it impossible to really critically evaluate the work!!!
  • Line 720: Results for microbiological and immunological outcomes are well written and short!

Discussion

  • Please use subheadings for discussion of different outcomes.
  • 7 pages of discussion is too long.

Supplement: - apical periodontitis is not periodontitis!

Author Response

Dear Reviewer,

Please see the attachment,

Thank you for the valuable points and your time to review our work.

Reviewer 2 Report

Dear authors,

I congratulate you for the effort made to carry out a comprehensive review to examine if probiotic supplements are associated with preventive and therapeutic benefits in terms of improvement of clinical, microbiological and immunological outcomes in patients with periodontal diseases. 

There are some major clarifications needed before I could review the manuscript further. 

  1. I see that there are multiple outcomes in your systematic review and meta-analysis. There are issues with multiplicity in a systematic review.

It is well understood that there are clinical trials that have more than one primary outcome and response to treatment may be measured in a variety of ways. Collating large amounts of data is not a problem but analyzing many outcomes is problematic when the conclusion that an intervention is beneficial is made if any outcome reaches statistical significance. One promising approach to deal with multiple outcomes in systematic reviews is given by multivariate meta-analysis, which has the advantage of providing a complete and concise description of all data simultaneously rather than a number of univariate meta-analyses separately for each outcome. To avoid multiple testing is to specify, a priori, which comparisons are of primary interest in the protocol for the review. This could be better understood if a clear/precise PICO question is presented. In the case of the current systematic review, I wonder what could be the PICO question OR could there be multiple PICO questions?

2. The next important issue; although Figure 2 claims to be following the PRISMA guidelines to report the flow of the study, it is far from it. Please refer to the following website to download the PRISMA flow diagram. Just mentioning the number of studies included or excluded without specifying the reasons is not enough (Line 319 onwards in the Result section).

http://www.prisma-statement.org/

3. Comprehensive review claim: The following are minimum requirements to claim a systematic review to be comprehensive. 

Please refer: https://www.cambridge.org/core/services/aop-cambridge-core/content/view/S1049023X16000649

1. Justification for the review.

An Introduction section should highlight the problem being reviewed and justify why it needs to be addressed.

2. Precise review Objective.

The question to be answered by the review should be unambiguous and clearly stated at the end of the Introduction section.

3. Statement of the inclusion and exclusion criteria for the studies entered into the review.

The inclusion criteria for literature placed into the review should include, at a minimum, an appropriate study design, specific populations, interventions, is of current practice, and outcomes of interest.

4. An explicit literature search should be defined and employed prior to beginning the literature search.

Search terms, language restrictions, databases, types of publications, study methods acceptable, and years to search should be determined for literature review.

5. Describe efforts to decrease selection bias.

Key variables that will be determined must be defined prior to the literature search. Optimally more than one reviewer reviews each literature reference and inter-rater reliability is determined and reported. A standardized review form with pre-review definitions should be used.

6. A substantial effort to identify all applicable literature as defined by the review inclusion criteria.

All applicable databases should be searched along with a secondary review of bibliographies of original review papers. For some types of reviews, a systematic search of “grey-literature” should be conducted using appropriate Internet search engines. Often an academic librarian can help.

7. Rank (weigh) the quality of the literature reviewed and assign higher significance for higher quality research. What funding sources were used for any literature included?

Randomized controlled trials and prospective studies are usually of higher quality than retrospective medical record reviews. Explicit criteria for ranking the validity of reviewed literature should be developed reported and summarized. A research funding source may subtlety bias the outcome.

8. Studies included should be suitable for answering the review Objective.

This is usually accomplished by providing a table or chart of the studies included in the review with information on author(s), study objective, study design, interventions, sample size, population studied, time frame of study, missing data, limitations, outcome measures, and final results.

9. Objectively report the Results of the review.

Provide an objective report of the findings; save interpretation for the Discussion section of the manuscript. A table or chart may be appropriate.

10. Discuss the Results of the review; focus interpretation of the review on the material presented.

The Discussion allows for an explanation of any findings relative to other published reports or similar topics.

11. Provide detailed Limitations experienced with the review.

Recognition of the Limitations of the review is important for readers and other researchers to allow for interpretation of review Results.

12. Provide a concise Conclusion.

A concise Conclusion that describes the primary review findings relative to the review Objective should be provided in a few sentences.

I look forward to your response.

Best wishes.

Author Response

Dear reviewer,

Thank you for the valuable points and your time to review our work.

Round 2

Reviewer 2 Report

The authors have addressed my comments. 

I have no further comments on this manuscript.

Author Response

Thank you so much for your time to review our work.